# The Role of the Molecular Hydrogen Formation in the Process of Metal-Ion Reduction on Multicrystalline Silicon in a Hydrofluoric Acid Matrix

**DOI:** 10.3390/nano11040982

**Published:** 2021-04-11

**Authors:** Stefan Schönekerl, Jörg Acker

**Affiliations:** Department of Physical Chemistry, Faculty of Environment and Natural Sciences, Brandenburg University of Technology Cottbus-Senftenberg, Universitätsplatz 1, 01968 Senftenberg, Germany; joerg.acker@b-tu.de

**Keywords:** molecular hydrogen, metal deposition, silicon dissolution, hydrofluoric acid, reaction process, valence transfer

## Abstract

Metal deposition on silicon in hydrofluoric acid (HF) solutions is a well-established process for the surface patterning of silicon. The reactions behind this process, especially the formation or the absence of molecular hydrogen (H_2_), are controversially discussed in the literature. In this study, several batch experiments with Ag^+^, Cu^2+^, AuCl_4_^−^ and PtCl_6_^2−^ in HF matrix and multicrystalline silicon were performed. The stoichiometric amounts of the metal depositions, the silicon dissolution and the molecular hydrogen formation were determined analytically. Based on these data and theoretical considerations of the valence transfer, four reasons for the formation of H_2_ could be identified. First, H_2_ is generated in a consecutive reaction after a monovalent hole transfer (h^+^) to a Si–Si bond. Second, H_2_ is produced due to a monovalent hole transfer to the Si–H bonds. Third, H_2_ occurs if Si–Si back bonds of the hydrogen-terminated silicon are attacked by Cu^2+^ reduction resulting in the intermediate species HSiF_3_, which is further degraded to H_2_ and SiF_6_^2−^. The fourth H_2_-forming reaction reduces oxonium ions (H_3_O^+^) on the silver/, copper/ and gold/silicon contacts via monovalent hole transfer to silicon. In the case of (cumulative) even-numbered valence transfers to silicon, no H_2_ is produced. The formation of H_2_ also fails to appear if the equilibrium potential of the 2H_3_O^+^/H_2_ half-cell does not reach the energetic level of the valence bands of the bulk or hydrogen-terminated silicon. Non-hydrogen-forming reactions in silver, copper and gold deposition always occur with at least one H_2_-forming process. The PtCl_6_^2−^ reduction to Pt proceeds exclusively via even-numbered valence transfers to silicon. This also applies to the reaction of H_3_O^+^ at the platinum/silicon contact. Consequently, no H_2_ is formed during platinum deposition.

## 1. Introduction

Silicon has a key role in various applications, for instance, in photovoltaics [1], sensors [2] or batteries [3]. In wet-chemical processing, the formation of special silicon surface modifications, e.g., helical nanopores [4], zigzag nanowires [5] or porous silicon [6], is of great importance to change the optical and electrical properties of the semiconductor. To control the fabrication processes, thin metal films are previously deposited on the silicon surface to catalyze the consecutive etching step initiated by various oxidants (e.g., H_2_O_2_) in the hydrofluoric acid (HF) matrix [7]. There are several options to create the metal coating. It can be realized by atomic layer deposition [8,9], several chemical or physical deposition techniques like spin coating, sputtering or electroless metal deposition [10]. The latter method is characterized by its simplicity, rapidity, versatility and scalability [7].

Despite decades of research, the reaction process of electroless metal deposition is not completely understood. This process is a redox reaction in which precious metal cations are reduced to a metal state and deposited as a film on the silicon surface (cathodic reaction) [11,12,13,14,15,16] without forming covalent bonds [15]. On the anodic side, the silicon is oxidized and dissolved in the HF matrix into the formal end product SiF_6_^2−^ [17].

There are different perceptions in the literature regarding whether or why the metal deposition process is accompanied by the formation of molecular hydrogen. The hypothesis is proposed in several studies that the occurrence of molecular hydrogen is due to a cathodic reduction of hydrogen ions (H^+^) coexisting with precious metal-ion reduction, either via electron uptake (e^−^) through the silicon conduction band or via hole transfer (h^+^) to the silicon valence band (Equation (1)) [18,19,20,21,22,23,24,25]:(1)$$2{\;H}^{+}+2{\;e}^{-}\to H_{2}\;\mathrm{or}\;2{\;H}^{+}\to H_{2}+2{\;h}^{+}$$

The cathodic reaction in our previous study [26] was interpreted as the reduction of the oxonium ion (H_3_O^+^) or the ion pair H_3_O^+^ F^−^ to molecular hydrogen at the metal/silicon contact. According to this model, the process occurs via the intermediate formation of hydrogen radicals (∙H), hydroxyl radicals (∙OH), water (H_2_O) and fluoride (F^−^). The valence exchange proceeds as a hole transfer from the oxonium ion to the valence band of silicon (Equation (2)):(2)$$\begin{array}{c} H_{3}O^{+}F^{-}\to 2\cdot H+\cdot \mathrm{OH}+F^{-}+{\;h}^{+}⇌\;\cdot H+H_{2}O+F^{-}+h^{+}\\ ⇌0.5{\;H}_{2}+H_{2}O\;+F^{-}+h^{+} \end{array}$$

The basic requirement for this process is the metal deposition on the silicon after the initial metal-ion reduction, i.e., forming a metal/silicon contact (Figure 1a,b) [26]. The Fermi energy equilibration (*E_F_*) at the metal/silicon contact occurs close to the energetic level of the standard hydrogen half-cell [13,27] at *E* (2H_3_O^+^/H_2_) = 0 V vs. the standard hydrogen electrode (SHE) [28]. Furthermore, the valence (*E_V_*) and the conduction band (*E_C_*) bending of the bulk silicon [19,24,29,30,31,32] and that of the hydrogen-terminated silicon results in the order of the difference of the work functions of the respective metal (*Φ_Me_*) and that of the hydrogen-terminated silicon (*Φ_Si-Hx_*) (Figure 1b) [26]. Depending on the current density *J* of the metal ions on the metal/silicon surface, either an underpotential or an overvoltage of the hydrogen electrode is established on the metal surface [33]. The lower the metal ion concentration is, the higher the equilibrium potential of the 2H_3_O^+^/H_2_ half-cell (*Eq. Pot.* (2H_3_O^+^/H_2_)) [33]. As a precondition for oxonium-ion reduction according to the scheme of Equation (2), the equilibrium potential of the 2H_3_O^+^/H_2_ half-cell must be at least equal to the energetic level of the valence band of the bulk silicon (Figure 1b). Accordingly, the initial molality of Ag^+^ = 3.3 × 10^−4^ mol∙kg^−1^, Cu^2+^ = 2.5 × 10^−4^ mol∙kg^−1^, AuCl_4_^−^ = 1.5 × 10^−4^ mol∙kg^−1^, or PtCl_6_^2−^ = 0.6 × 10^−4^ mol∙kg^−1^ must not be exceeded, otherwise the process does not take place (Figure 1c) [19].

In the literature, it is further assumed that molecular hydrogen can also be formed at the anodic side. Turner [17] and Memming and Schwandt [34] postulated that when the current density on the silicon falls below a certain level depending on the metal [35], the bulk silicon formally goes into the HF solution as Si^2+^ after the initial divalent valence transfer. The unstable Si^2+^ species [17,34] is subsequently converted by an H(+I)-based oxidant to the Si^4+^ state (SiF_6_^2−^) [17,34] with the formation of H_2_. There are different assumptions as to whether the oxidant is H^+^ [36,37,38,39], H_2_O [34], HF [17,34,40] or the HF dissociation product HF_2_^−^ [21,41,42,43,44,45,46,47,48,49] (Equation (3)):(3)$$\begin{array}{c} {\mathrm{Si}}^{2+}+\left(2{\;H}^{+}+6{\;F}^{-}\right)\;\mathrm{or}\;\left(2{\;H}_{2}O+6{\;F}^{-}\right)\;\mathrm{or}\;\left(6\;\mathrm{HF}\right)\;\mathrm{or}\;\left(4{\;\mathrm{HF}}_{2}{}^{-}\right)\\ \;\;\;\;\;\;\;\;\;\to {\mathrm{SiF}}_{6}{}^{2-}+H_{2}+\left(-\right)\;\mathrm{or}\;\left(2{\;\mathrm{OH}}^{-}\right)\;\mathrm{or}\;\left(4{\;H}^{+}\right)\;\mathrm{or}\;\left(2\;\mathrm{HF}\right) \end{array}$$

According to the most prominent theory, the course of the reaction changes from the divalent to the tetravalent mechanism when a certain current density is exceeded [17,34]. The bulk silicon is oxidized to Si^4+^ and, subsequently, converted to the formal end product SiF_6_^2−^ in the presence of F^−^ [11,12,29,50,51,52,53,54], HF [15,21,34,41,44,55] or HF_2_^−^ [21,44] with the intermediate formation of SiF_4_ [34,40] without the formation of molecular hydrogen (Equation (4)):
(4)$${\mathrm{Si}}^{4+}+\left(6{\;F}^{-}\right)\;\mathrm{or}\;\left(6\;\mathrm{HF}\right)\;\mathrm{or}\;\left(3{\;\mathrm{HF}}_{2}{}^{-}\right)\to {\mathrm{SiF}}_{6}{}^{2-}+\left(-\right)\;\mathrm{or}\;\left(6{\;H}^{+}\right)\;\mathrm{or}\;\left(6{\;H}^{+}\right)$$

It was found in our previous study [26] that stoichiometric ratios of metal deposition on silicon change gradually as a function of the initial metal ion molality. The ratios increased with the increasing metal ion concentration from just above 0:1 mol:mol on metal-dependent maxima, indicating a rising proportion of silicon oxidation up to the oxidation state of +IV by metal-ion reduction.

Furthermore, it was concluded that the kinetics of metal deposition became maximal when the redox half-cell of the metal ions (Me^z+^/Me) was sufficiently strong to reach both the valence’s energetic level band of the bulk silicon and that of the hydrogen-terminated silicon. The metal ion concentration thresholds, matching the energetic level of the hydrogen-terminated silicon, depending on the amount of band bending at the metal/silicon contact [26].

Some studies have proposed that the oxidation of the hydrogen-terminated silicon, i.e., that of the ≡Si–H, =Si=H_2_ or –Si≡H_3_ groups [56], is also associated with the formation of molecular hydrogen. According to the theory of Lehmann and Gösele [57] and Bertagna et al. [58], the molecular hydrogen is formed by the Si–H bond breaking as a result of the valence exchange with the oxidant, basically, as shown in Equation (5):(5)$$=\mathrm{Si}=H_{2}+2{\;h}^{+}\to {\;\mathrm{Si}}^{2+}+H_{2}$$

Miura et al. [59] and Boonekamp et al. [60] have demonstrated elsewhere that the Si–H bonds are corrodible by H_2_O without the presence of HF, and, consequently, molecular hydrogen is formed.

According to the models of Gerischer et al. [61], Kooij and Vanmaekelbergh [62], Kolasinski [63], and Stumper and Peter [36], the Si–H bonds are not attacked, but the Si–Si back bonds of the hydrogen-terminated silicon groups are. Subsequently, F^−^ forms covalent bonds with the oxidized silicon, leading to the formation of HSiF_3_. This species migrates into the HF solution, where it is converted to SiF_6_^2−^ and H_2_ under different assumptions about the reaction pathway (e.g., Equation (6) according to [63]):(6)$${\mathrm{HSiF}}_{3}\;+\;\mathrm{HF}\;+\;2{\;F}^{-}\to {\mathrm{SiF}}_{6}{}^{2-}+H_{2}$$

Contradicting these theses, Tsuboi et al. [14] and Ogata et al. [20] consider the reaction at the hydrogen-terminated silicon without the formation of molecular hydrogen, according to Equation (7):(7)$$\begin{array}{c} \mathrm{Si}-H_{x}+\left(4+x\right){\;h}^{+}+2{\;H}_{2}O\;\to {\;\mathrm{SiO}}_{2}+\left(4+x\right){\;H}^{+}\;\mathrm{and}\;{\mathrm{SiO}}_{2}+6\;\mathrm{HF}\to \\ {\mathrm{SiF}}_{6}{}^{2-}+2{\;H}^{+}+\;2{\;H}_{2}O \end{array}$$

All the theories presented on the origin of molecular hydrogen have the disadvantage of being proposed without an analytical quantification of the molecular hydrogen formation. The aim of this study is to overcome this gap in understanding and to verify the theories based on different types of metal depositions on silicon. In addition to understanding the reaction processes between metal ions and silicon, the issue of hydrogen formation is also of interest, as in the metal-catalyzed degradation of halogenates (e.g., ClO_3_^−^) with molecular hydrogen [64] or the hydrogen adsorption on metals [65] or the hydrogen storage in metal–organic frameworks [66].

Several series of experiments have been carried out based on HF solutions with Ag^+^, Cu^2+^, AuCl_4_^−^ or PtCl_6_^2−^ and multicrystalline silicon. The metal ions were chosen because they are commonly used in the metal-assisted etching of silicon [7,8,9,10,11,12,13,14,15,16,18,19,20,21,22,23,24,26,27,29,30,31,32,37,38,39,40,41,42,43,44,45,46,47,48,49,50,51,52,53,54,63,67,68,69,70,71,72,73,74,75,76,77]. Furthermore, the selected metal ion species are characterized by their different charge numbers (Ag(I), Cu(II), Au(III) and Pt(IV)). This aspect is of particular interest concerning molecular hydrogen formation, which is discussed subsequently. The metal ions’ initial concentrations were varied, and the concentration of HF in the silver and copper deposition experiments was also modified. The amounts of the dissolved metal ion and silicon species were determined analytically at the beginning, and the end of each experiment and the amount of molecular hydrogen transferred to the gas phase was monitored. The stoichiometric ratios of the metal deposition, silicon dissolution and molecular hydrogen formation derived from the findings were interpreted in terms of theoretical reaction processes.

## 2. Materials and Methods

Common procedure: A 150 mL low-density polyethylene vessel (VITLAB, Großostheim, Germany) was used for all experiments. In each case, a solution with a mass of *m_Sol_* = 0.060 ± 0.002 kg was transferred into this vessel. The solution consisted of ultrapure water (18 MΩ resistance, PURELAB^®^ flex, ELGA VEOLIA, Celle, Germany), an admixture of 40% HF (*w**/w*, p.A. quality, EMSURE^®^, Merck KGaA, Darmstadt, Germany) and a specific addition from a metal ion stock solution. The solutions were continuously mixed at *T* = 295 +/− 2 K using a polytetrafluoroethylene (PTFE)-coated magnetic stir bar (Alnico, 8 × 20 mm, VWR collection, Radnor, PA, USA) and a stir plate (RT 10 power, IKA^®^, Staufen, Germany) at a rotation speed of about 125 rpm. Subsequently, the solution was sampled for the analytical determination of the initial molalities of the dissolved metal and silicon species using inductively coupled plasma atomic emission spectroscopy (ICP–OES) (*b* Me (*diss., t* = 0 s), Me = Ag, Cu, Au, Pt and *b* Si (*diss., t* = 0 s)). Before the closure of the vessel, ten multicrystalline silicon wafers (*m_Si_* = 0.92 ± 0.03 g, edge length of 15 mm, the thickness of 0.175 mm, SCHOTT Solar), pretreated with 1% HF (*v**/v*), were attached to the inside of the GL45 cap (Bohlender, Grünsfeld, Germany) using a PTFE thread. The thread was fixed to one of the three GL-14 ports on the top of the GL 45 cap (Bohlender, Grünsfeld, Germany). Afterward, the solution was continuously flushed with an argon flow set to 45 ± 1 mL∙min^−1^ (gas flow meter rotameter Yokogawa, Ratingen, Germany, Ar 5.2, Air Products, Danbury, CT, USA) through the other two GL-14 ports via a PTFE hose line. The exhaust gas hose line was coupled with a gas phase mass spectrometer. After about 30 min of purging, the oxygen initially presents in the gas phase of the sample vessel was removed. The Si wafers were then transferred into the solution by the PTFE thread being pulled externally. While maintaining stirring and argon rinsing, the gas phase composition was continuously measured and recorded for *t* = 0 s to *t* = 3600 s for the Ag, Cu and Au deposition experiments and for *t* = 0 s to *t* = 14,400 s for the Pt deposition experiments. After reaching the time of *t* = 3600 s or *t* = 14,400 s, the vessel was opened, and the solution was sampled to determine the element concentrations.

Ag^+^ solutions: In the first step, an AgF/HF stock solution with a concentration of 2.9 mol∙kg^−1^ AgF/HF was prepared by intermediate preparation of Ag_2_O from an AgNO_3_ solution (≥99%, ACS reagent, Sigma-Aldrich, St. Louis, MO, USA) and the addition of 25% NaOH solution (25%, *w/w*, extra pure, Carl Roth, Karlsruhe, Germany), followed by filtration, ultrapure water rinsing, drying and subsequent HF dissolution. Based on this metal stock solution and the 40% HF stock solution (*w**/w*), eight solutions were prepared with a HF molality of 0.056 ± 0.003 mol∙kg^−1^ with initial Ag^+^ molalities of *b* Ag (*diss., t* = 0 s) = 1.4 × 10^−6^ to 2.6 × 10^−2^ mol∙kg^−1^, nine solutions with a HF molality of 0.280 ± 0.016 mol∙kg^−1^ with initial Ag^+^ molalities of *b* Ag (*diss., t* = 0 s) = 1.4 × 10^−6^ to 1.2 × 10^−1^ mol∙kg^−1^ and nine solutions with a HF molality of 1.339 ± 0.019 mol∙kg^−1^ with initial Ag^+^ molalities of *b* Ag (*diss., t* = 0 s) = 1.9 × 10^−6^ to 1.5 × 10^−1^ mol∙kg^−1^.

Cu^2+^ solutions: Regarding the preparation of the Cu^2+^ solutions, CuSO_4_·5H_2_O (≥98%, ACS reagent, Sigma-Aldrich, St. Louis, MO, USA) was used. Eleven solutions were prepared with a HF molality of 0.060 ± 0.001 mol∙kg^−1^ and initial Cu^2+^ molalities of *b* Cu (*diss.*, *t* = 0 s) = 1.2 × 10^−4^ to 6.2 × 10^−2^ mol∙kg^−1^, 18 solutions with a HF molality of 0.305 ± 0.005 mol∙kg^−1^ and initial Cu^2+^ molalities of *b* Cu (*diss.*, *t* = 0 s) = 1.5 × 10^−4^ to 7.7 × 10^−2^ mol∙kg^−1^ and 17 solutions with a HF molality of 1.502 ± 0.024 mol∙kg^−1^ and initial Cu^2+^ molalities of *b* Cu (*diss.*, *t* = 0 s) = 7.6 × 10^−4^ to 1.5 × 10^−1^ mol∙kg^−1^.

AuCl_4_^−^ solutions: The Au(III) solutions were prepared using a 40–44% HAuCl_4_ solution (*w/w*, ≥98%, ACS reagent, Sigma-Aldrich, St. Louis, MO, USA) with initial molalities of *b* Au (*diss.*, *t* = 0 s) = 4.8 × 10^−7^ to 7.6 × 10^−3^ mol∙kg^−1^ at an HF molality level of 1.333 ± 0.010 mol∙kg^−1^ in a total of ten batches.

PtCl_6_^2−^ solutions: Regarding the platinum deposition experiments, H_2_PtCl_6_·6H_2_O (≥37.5% Pt base, ACS reagent, Sigma-Aldrich, St. Louis, MO, USA) was selected. Thus, a total of nine solutions were prepared with initial Pt(IV) molalities ranging from *b* Pt (*diss.*, *t* = 0 s) = 4.5 × 10^−7^ to 1.0 × 10^−2^ mol∙kg^−1^ with an HF molality level of 1.486 ± 0.016 mol∙kg^−1^.

Determination of the molalities of the elements in the HF solutions: The HF solutions were sampled at the beginning (*t* = 0 s) and after *t* = 3600 s in the case of the silver, copper and gold deposition experiments or after *t* = 14,400 s in the case of the platinum deposition experiments. The subsequent analysis of the samples was performed by the ICP–OES system iCAP 6500 Duo (Thermo Fisher, Waltham, MA, USA). Herewith, the spectral emissions of the elements Ag (*λ* = 243.7 nm, 328.0 nm, 338.2 nm), Cu (*λ =* 224.7 nm, 324.7 nm, 327.3 nm), Au (*λ* = 201.2 nm, 242.7 nm, 267.5 nm), Pt (*λ* = 214.4 nm, 265.9 nm, 306.4 nm) and Si (*λ* = 212.4 nm, 251.6 nm, 252.8 nm) were measured axially to the torch. The molalities of the elements *b* (mol∙kg^−1^) inclusive of its analytical errors (+/− confidence interval) were determined by system calibrating via measuring the dilution series of 1000 mol∙kg^−1^ standard solutions (Cu, Ag, Au, or Pt and Si, CertiPUR^®^, Merck KGaA, Darmstadt, Germany). The stoichiometric amount of metal cation reduction to a metal state Δ*n* Me (*diss.*, *t* = 3600 s or 14,400 s) (Me = Ag, Cu, Au, or Pt) and the amount of silicon dissolution Δ*n* Si (*diss.*, *t* = 3600 s or 14,400 s) were calculated from the difference in the molalities of the elements from the sampling at *t* = 0 s and *t* = 3600 s or 14,400 s multiplied by the mass of the solutions *m_sol_*.

Determination of the molecular hydrogen in the gas phase: The GSD 320 O2C OmniStar gas phase mass spectrometer (Pfeiffer Vacuum, Aßlar, Germany) was coupled to the sampling port of the exhaust hose line. The system extracted 2 cm^3^∙min^−1^ of the exhaust gas continuously via a polyether ether ketone hose line heated to *T* = 403 K. The exhaust gas was analyzed by Faraday detection. The molecular hydrogen was determined routinely. The calibration was performed by the preparation of molecular hydrogen from the reaction of granulated zinc (3 mm in diameter, p.A., Merck KGaA, Darmstadt, Germany) with 0.06 kg 20% HCl (*w*/*w*, p.A., Merck KGaA, Darmstadt, Germany), according to the reaction scheme Zn*_(s)_ + 2* HCl*_(aq)_ →* ZnCl_2*(aq)*_
*+* H_2*(g)*_. An individual calibration based on 8 to 25 different molar amounts of zinc from about 5.0 × 10^−6^ to 1.2 × 10^−3^ mol was performed for each experimental series. The gas-phase above the hydrochloric acid matrix was continuously flushed with an argon flow of 45 ± 1 mL∙min^−1^, similar to the metal deposition experiments, and the gas mixture was extracted continuously in a partial flow through the measuring instrument. The ion current generated in the detector at the mass of 2 amu was recorded with a time interval of about 0.9 s. The surface integrals were calculated based on the measurement signal and the time until the molecular hydrogen was removed completely from the vessel. In the context of calibration, these values were related to the respective molar amounts of molecular hydrogen, which are in a 1:1 mol:mol stoichiometric ratio to the amount of zinc. This method follows Rietig et al. [78]. The amount of molecular hydrogen Δ*n* H_2_ (*g, t =* 3600 s) or Δ*n* H_2_ (*g*, *t* = 14,400 s) in the metal deposition experiments was calculated inversely from the integrals of the ion current at 2 amu of *t* = 0 s to *t* = 3600 s (Ag, Cu, Au) or to *t* = 14,400 s (Pt) via the calibration (+/− confidence interval). The statistical detection limit for molecular hydrogen was about 1.8 × 10^−6^ mol.

## 3. Results and Discussion

### 3.1. Silver Deposition onto Multicrystalline Silicon

Silver deposition was studied within 26 single experiments with initial Ag^+^ molalities ranging from 1.2 × 10^−6^ to 1.5 × 10^−1^ mol∙kg^−1^ at three different HF molality levels (0.056 ± 0.003 mol∙kg^−1^, 0.280 ± 0.016 mol∙kg^−1^ and 1.339 ± 0.019 mol∙kg^−1^). Based on Nernst’s equation, the standard potential of the Ag^+^/Ag half-cell of 0.779 V vs. SHE (referenced to AgF) [28] and the calculation of the activity coefficients of the Ag^+^ ions according to Bromley’s equations [79], the initial redox strengths for the Ag^+^/Ag half-cell were determined to be between 0.43 and 0.72 V vs. SHE (Figure 2a).

It is assumed from the findings of our previous study [19] that one up to four holes can be transferred to the valence band of the bulk silicon during the reduction of Ag^+^ to Ag (Equation (8)):(8)$$n{\;\mathrm{Ag}}^{+}\to n\;\mathrm{Ag}\;+\;n{\;h}^{+}\;\mathrm{and}\;\mathrm{Si}\;+\;n{\;h}^{+}\to {\mathrm{Si}}^{n+}\;\mathrm{with}\;1\;\leq n\leq 4$$

Kinetic considerations of the silver deposition in the same study led to the conclusion that valence transfer also occurs via the valence band of hydrogen-terminated silicon when the redox strength of the Ag^+^/Ag half-cell exceeds the amount of 0.65 V vs. SHE (corresponding to *b* Ag (*diss.*, *t* = 0 s) = 5.5 × 10^−3^ mol∙kg^−1^) [26].

The redox strength of the Ag^+^/Ag half-cell must be at least at the energetic level of the valence bands of the bulk silicon or hydrogen-terminated silicon at the silver/silicon contact to enable the valence transfer. The energetic level of the valence band of the bulk silicon without metal contact in the vacuum state is *E_V_* (Si_bulk_) = −0.41 eV [26,80,81] and that of the hydrogen-terminated silicon is *E_V_* (Si-H_x_) = −0.68 eV [19,82] (Figure 2a). After the initial silver/silicon contacting, band bending of the conduction and the valence bands of silicon occurs due to Fermi energy alignment at the contact. The amount of band bending is small due to the small average differences in the work functions of silver and the hydrogen-terminated silicon [83,84,85,86,87,88]. Based on the Ag^+^/Ag half-cell threshold of *E* (Ag^+^/Ag) = 0.65 V vs. SHE experimentally determined in [26], an energetic level of the valence band of hydrogen-terminated silicon at the silver/silicon contact of *E_V_* (Si-H_x_) ≈ −0.65 eV can be assumed. This results in a band bending in the amount of +0.03 eV (*E_V_* (Si-H_x_) = −0.68 eV + 0.03 eV = −0.65 eV, Figure 2a). If the redox level of the Ag^+^/Ag half-cell is below this threshold, the valence transfer occurs via the valence band of the bulk silicon. If an identical magnitude of band bending of +0.03 eV is assumed for the valence band of bulk silicon, its energetic level at the silver/silicon contact is *E_V_* (Si_bulk_) = −0.38 eV (= −0.41 eV + 0.03 eV, Figure 2a). This means that the Ag^+^/Ag half-cell must exceed a redox level of *E* (Ag^+^/Ag) = 0.38 V vs. SHE to initiate the valence transfer to the bulk silicon. The *E* (Ag^+^/Ag) was about 0.05 V above this threshold for the lowest concentrated Ag^+^ solution. Consequently, the valence transfer from the Ag^+^ reduction to the bulk silicon could occur in all solutions.

The cathodic process of the oxonium-ion reduction is more limited. Based on the previous study, the initial Ag^+^ molality of *b* Ag (*diss.*, *t* = 0 s) = 3.3 × 10^−4^ mol∙kg^−1^ was identified as the threshold up to which the oxonium-ion reduction can coexist with the Ag^+^ reduction after the initial silver nucleation [26]. Below this threshold, the equilibrium potential of the 2H_3_O^+^/H_2_ half-cell is sufficiently strong that monovalent hole transfer via the silver/silicon contact can occur via the valence band of the bulk silicon (Equation (2)). The current density at the threshold depending on the Ag^+^ molality at the silver/silicon wafer surface *J* (Ag^+^) is about 2.1 × 10^−2^ mA∙cm^−2^ based on the calculation method in [26]. According to the findings of Hunt et al. [26], this result is equal to an equilibrium potential of the 2H_3_O^+^/H_2_ half-cell of *Eq. Pot.* (2H_3_O^+^/H_2_) = +0.38 V vs. SHE. This value corresponds to the postulated energetic level of the top of the valence band of the bulk silicon at the silver/silicon contact of *E_V_* (Si_bulk_) = −0.38 eV. The current density *J* (Ag^+^) for the lowest concentrated Ag^+^ solution is 4.2 × 10^−5^ mA∙cm^−2^, resulting in an equilibrium potential of *Eq. Pot.* (2H_3_O^+^/H_2_) = +0.57 V vs. SHE [33] (Figure 2a).

Based on these considerations, the analytical findings for the amounts of silver deposition (Δ*n* Ag (*diss.*, *t* = 3600 s)), silicon dissolution (Δ*n* Si (*diss.*, *t* = 3600 s)) and molecular hydrogen formation (Δ*n* H_2_ (*g*, *t* = 3600 s)) shown in Figure 2b–d as a function of the initial Ag^+^ molality (*b* Ag (*diss.*, *t* = 0 s)) were color classified. The results in which the oxonium-ion reduction participated in the oxidation of the bulk silicon are marked in red. The analytical findings in which the Ag^+^/Ag half-cell interacted with the bulk silicon without the involvement of the 2H_3_O^+^/H_2_ half-cell were labeled in green. The black highlighted results symbolize the experiments in which the Ag^+^/Ag half-cell was strong enough to react with the hydrogen-terminated silicon. The black and white shaded symbols index the same reaction class but with kinetically limited silicon dissolution. According to the results of our previous study [19], time-delayed silicon dissolution occurs if the initial activities of the anionic HF dissociation products F^−^, HF_2_^−^ and H_2_F_3_^−^ (*a* F^−^, HF_2_^−^, H_2_F_3_^−^) are cumulatively below a ratio of 6:1 mol∙kg^−1^:mol∙kg^−1^ relative to the molality of the silicon dissolved.

Figure 2b illustrates that within the period of *t* = 3600 s, the amount of Ag^+^ was reduced to metallic silver by an average of 97% in all 26 individual experiments. Thus, the product of the initial Ag^+^ molality and the mass of solution (*b* Ag (*diss.*, *t* = 0 s) × *m_Sol_*) is almost equal to the determined amount of Δ*n* Ag (*diss.*, *t* = 3600 s).

The amount of silicon dissolution (Δ*n* Si (*diss.*, *t* = 3600 s)) is plotted in relation to *b* Ag (*diss., t* = 0 s) in Figure 2c. A significant change in the slope of silicon dissolution can be noticed at the proposed threshold of oxonium-ion reduction at *b* Ag (*diss., t* = 0 s) = 3.3 × 10^−4^ mol∙kg^−1^ [26]. The oxidation of silicon to the left of the threshold is induced by the reduction of Ag^+^ and H_3_O^+^, and to the right, by the reduction of Ag^+^ without the participation of H_3_O^+^.

Figure 2d shows the amounts of molecular hydrogen formation (Δ*n* H_2_ (*g, t* = 3600 s)). Molecular hydrogen was produced in all 26 individual experiments. The lowest amount of molecular hydrogen formation was observed in the experiment with the lowest initial Ag^+^ molality of *b* Ag (*diss., t* = 0 s) = 1.2 × 10^−6^ mol∙kg^−1^. The amount of Δ*n* H_2_
*(g, t* = 3600 s) was ≈1.8 × 10^−6^ mol, narrowly above the detection limit of the method selected. The formation of molecular hydrogen in the oxonium-ion reduction range increases linearly with the increasing initial Ag^+^ molality, reaching an extrapolated level of Δ*n* H_2_ (*g, t* = 3600 s) of ≈6.3 × 10^−6^ mol at the threshold of oxonium-ion reduction at *b* Ag (*diss., t* = 0 s) = 3.3 × 10^−4^ mol∙kg^−1^.

The molecular hydrogen formation for the solutions with an initial HF molality of 0.056 ± 0.003 mol∙kg^−1^ increased to Δ*n* H_2_ (*g, t* = 3600 s) ≈ 9.0 × 10^−6^ mol up to *b* Ag (*diss., t* = 0 s) = 5.5 × 10^−3^ mol∙kg^−1^. The oxidation of the hydrogen-terminated silicon is initiated above the level of *b* Ag (*diss., t* = 0 s) = 5.5 × 10^−3^ mol∙kg^−1^ [26]. Within the two experimental series with HF molalities of 0.280 ± 0.016 mol∙kg^−1^ and 1.339 ± 0.019 mol∙kg^−1^, Δ*n* H_2_ (*g, t* = 3600 s) was ≈3.0 × 10^−5^ mol at this threshold. The amount of molecular hydrogen formation for the solutions with a HF molality of 0.056 ± 0.003 mol∙kg^−1^ increased up to a maximum of Δ*n* H_2_ (*g, t* = 3600 s) ≈ 1.1 × 10^−5^ mol at *b* Ag (*diss., t* = 0 s) = 1.3 × 10^−2^ mol∙kg^−1^ and then decreased with increasing initial Ag^+^ molality. At *b* Ag (*diss., t* = 0 s) = 2.6 × 10^−2^ mol∙kg^−1^, Δ*n* H_2_ (*g, t* = 3600 s) was ≈5.4 × 10^−6^ mol. The molecular hydrogen formation of the experiments with a HF molality of 0.280 ± 0.016 mol∙kg^−1^ reached a level of Δ*n* H_2_ (*g, t* = 3600 s) = (5.4 ± 0.7) × 10^−5^ mol above an initial Ag^+^ molality of 9.0 × 10^−3^ mol∙kg^−1^. A higher-level plateau of Δ*n* H_2_ (*g, t* = 3600 s) = (2.0 ± 0.1) × 10^−4^ mol was obtained at *b* Ag *(diss., t* = 0 s) ≥ 3.5 × 10^−2^ mol∙kg^−1^ for the solutions with an initial HF molality of 1.339 ± 0.019 mol∙kg^−1^.

Based on the analytically determined data presented in Figure 2b–d, the amounts of silver deposition were stoichiometrically related to the amounts of the silicon dissolution (Δ*n* Ag*:*Δ*n* Si (*diss., t* = 3600 s)) as a function of the initial Ag^+^ molality *b* Ag (*diss., t* = 0 s) in Figure 2e. The ratios of the molecular hydrogen formation and the silicon dissolution (Δ*n* H_2_*:*Δ*n* Si (*g* or *diss., t* = 3600 s) were determined analogously and are shown in the same relation to *b* Ag (*diss., t* = 0 s) in Figure 2f. The indicated error bars result from the confidence intervals of the values measured.

Comparing the silver deposition and the silicon dissolution, it can be seen that the Δ*n* Ag*:*Δ*n* Si (*diss., t* = 3600 s) ratio is about 1:35 mol:mol at the lowest selected initial Ag^+^ molality of *b* Ag (*diss., t* = 0 s) = 1.2 × 10^−6^ mol∙kg^−1^. As the initial Ag^+^ molality increases, the Δ*n* Ag*:*Δ*n* Si *(diss., t* = 3600 s) ratios increase sigmoidally and reach a maximum above the threshold of *b* Ag (*diss., t* = 0 s) ≈ 5.5 × 10^−3^ mol∙kg^−1^ between Δ*n* Ag*:*Δ*n* Si (*diss., t* = 3600 s) ≈ 3.33:1 mol:mol and 4.28:1 mol:mol. All stoichiometric ratios shown in Figure 2e are to be interpreted as time-weighted averages of varying stoichiometric ratios between *t* = 0 s and *t* = 3600 s. It was demonstrated in our previous study that the stoichiometric ratios change significantly during the deposition process [26]. The maxima and minima of these stoichiometric ratios determined in [26] are delimited by the sigmoid functions plotted in Figure 2e.

By contrast, the course of the ratios of the molecular hydrogen formation and the silicon dissolution is inversely sigmoid (Figure 2f). The Δ*n* H_2_*:*Δ*n* Si (*g* or *diss., t* = 3600 s) ratios in the range of oxonium-ion reduction are between (0.75 ± 0.45):1 and (0.71 ± 0.45):1 mol:mol. The Δ*n* H_2_*:*Δ*n* Si (*g* or *diss., t* = 3600 s) ratios decrease to a level of (0.46 ± 0.35):1 mol:mol up to the threshold of *b* Ag (*diss., t* = 0 s) ≈ 5.5 × 10^−3^ mol∙kg^−1^, where the oxidative attack on the hydrogen-terminated silicon begins. The Δ*n* H_2_*:*Δ*n* Si (*g* or *diss., t* = 3600 s) ratio is about (0.05 ± 0.04):1 mol:mol in the experiment with the maximum initial Ag^+^ molality of *b* Ag (*diss., t* = 0 s) = 1.5 × 10^−1^ mol∙kg^−1^.

Possible reaction processes can be derived based on a combined view of the stoichiometric ratios of silver deposition, silicon dissolution and molecular hydrogen formation (Δ*n* Ag*:*Δ*n* Si*:*Δ*n* H_2_) and the energetic considerations visualized in Figure 2a. Consequently, four exemplary results were selected from Figure 2e,f and explained.

Example 1 (Ex. 1, highlighted in red), with its stoichiometric Δ*n* Ag*:*Δ*n* Si*:*Δ*n* H_2_ ratio of ≈0.32:1:0.83 mol:mol:mol, is averaged from three individual experiments at an initial average Ag^+^ molality of *b* Ag (*diss., t* = 0 s) ≈ 1.5 × 10^−6^ mol∙kg^−1^. Figure 3 illustrates the reaction processes proposed. There are two main reaction complexes. Figure 3a shows the initial silver deposition, and Figure 3b, the consecutive oxonium-ion reduction. The initial silver deposition starts with the sorption of Ag^+^ species at the hydrogen-terminated silicon surface. The following Ag^+^ reduction to Ag formally results in transferring one hole to one Si–Si bond of the bulk silicon located below the hydrogen-terminated silicon (Figure 3a-1. The average redox level of the Ag^+^/Ag half-cells is *E* (Ag^+^/Ag) = 0.43 V vs. SHE. Thus, the valence band of the bulk silicon is accessible (*E_V_* (Si_bulk_) = −0.41 eV, Figure 2a).

The Si–H bonds of the hydrogen-terminated silicon cannot be affected (*E_V_* (Si-H_x_) = −0.68 eV, Figure 2a). A Si^+^ intermediate is formed to which F^−^ (or HF_2_^−^ or H_2_F_3_^−^) binds due to the monovalent valence transfer to the Si–Si bond. A dangling bond (●Si) is produced on the other side (Equation (9)):(9)$${\mathrm{Ag}}^{+}+F^{-}+\left(\equiv \mathrm{Si}-\mathrm{Si}\equiv \right)\to \mathrm{Ag}\;+\;\left(\equiv \mathrm{SiF}\;+●\mathrm{Si}\equiv \right)$$

It can be assumed that either primary H_2_O and secondary F^−^ (Figure 3a-2–4 or HF react with this dangling bond to form a hydrogen radical (∙H) and a silanol group (Si–OH) and, subsequently or immediately, a Si–F bond (Figure 3a-5). The hydrogen radical generated binds to another hydrogen radical from a similar reaction in the immediate vicinity and enters the gas phase as molecular hydrogen (Equation (10)):(10)$$\equiv \mathrm{Si}●\;+H_{2}O\;+F^{-}\to \;\equiv \mathrm{Si}-\mathrm{OH}+F^{-}\;+\;\cdot H⇌\;\equiv \mathrm{Si}-F\;+{\;\mathrm{OH}}^{-}\;+0.5{\;H}_{2}$$

The formation of the Si–F bonds destabilizes the remaining three adjacent Si^+^–Si bonds of each of the two Si^+^ intermediates (Figure 3a-6). This facilitates the reaction of H_2_O with the Si^+^–Si bonds. The HF would also be conceivable as a reactive species. However, it was found in our previous study [26] that the kinetics of silicon dissolution did not depend on the molality of HF, as long as the anionic HF dissociation species F^−^, HF_2_^−^ and H_2_F_3_^−^ were not limited. From this observation and the fact that the molality of water is at least 40 times greater than the undissociated HF, the reaction with H_2_O is more likely. Therefore, it is assumed that the formation of a silanol group occurs on the side of the oxidized Si, with the transfer of one valence to the H(+I) of H_2_O. The resulting hydrogen radical binds with the opposite Si atom and creates a Si–H bond (Figure 3a-7, Equation (11)):(11)$$\left(\equiv \mathrm{Si}\cdot \cdot \mathrm{SiF}=\right)+H_{2}O\to \;\left(\equiv \mathrm{Si}-H\;+\mathrm{HO}-\mathrm{SiF}=\right)$$

The reaction continues at the remaining Si^2+^–Si bonds. The silanol groups are subsequently replaced by F^−^ and, finally, the formal end product SiF_6_^2−^ is formed (Figure 3a-7,8).

In summary, the reaction process proposed in Figure 3a results in a stoichiometric Δ*n* Ag*:*Δ*n* Si*:*Δ*n* H_2_ ratio of 0.5:1:0.25 mol:mol:mol. The reaction of the oxonium ion or the H_3_O F^−^ ion pair, respectively, must be considered to justify the stoichiometric Δ*n* Ag*:*Δ*n* Si*:*Δ*n* H_2_ ratio of the selected example 1 of ≈0.32:1:0.83 mol:mol:mol (Figure 3b).

If the initial Ag^+^ molality is below the threshold of *b* Ag (*diss., t* = 0 s) = 3.3 × 10^−4^ mol∙kg^−1^, the 2H_3_O^+^/H_2_ half-cell is strong enough to initiate a monovalent valence transfer via the silver/silicon contact to the bulk silicon valence band due to the hydrogen underpotential effect on the silver surface. The influence of the band bending mechanism at the silver/silicon contact is of minor importance (Figure 2a).

In accordance with Equation (2), a radical reaction of the oxonium ion with a Si–Si bond occurs with the formation of one hydrogen radical and water on the cathodic side. On the anodic side, one of the Si atoms of the Si–Si bond attacked is oxidized and will be bonded by F^−^ and a dangling bond is created the opposite. This dangling bond reacts consecutively with the water formed from the H_3_O^+^ reduction to form a silanol group and another hydrogen radical. Both hydrogen radicals subsequently combine to form molecular hydrogen (Equation (12), Figure 3b-1–3:(12)$$\left(\equiv \mathrm{Si}-\mathrm{Si}\equiv \right)+H_{3}O^{+}F^{-}\to \left(\equiv \mathrm{SiF}\;+●\mathrm{Si}\equiv \right)+\cdot H+H_{2}O\to \;\left(\equiv \mathrm{SiF}\;+\mathrm{OH}-\mathrm{Si}\right)\;+2\;\cdot H\;\;⇌H_{2}$$

This reaction step is followed by more reactions of the oxonium ion or the ion pair H_3_O^+^ F^−^ with the adjacent Si–Si bonds with a further formation of molecular hydrogen, silanol groups and, subsequently, Si–F groups and more dangling bonds. If the number of dangling bonds is even and they are opposite each other, they can combine to form new Si–Si bonds (Figure 3b-4–8. As a result of the reaction sequences outlined in Figure 3b, the Δ*n* Ag*:*Δ*n* Si*:*Δ*n* H_2_ ratio is 0:1:2 mol:mol:mol.

Summarizing the partial reactions of silver deposition (Figure 3a) and that of consecutive oxonium-ion reduction (Figure 3b), the overall reaction balance shown in Figure 3c is approximately obtained with the average Δ*n* Ag*:*Δ*n* Si*:*Δ*n* H_2_ ratio of ≈ 0.32:1:0.83 mol:mol:mol indexed in Figure 2e,f.

In conclusion, about 20% of the molecular hydrogen formation originates from the reaction of water with dangling bonds, which is consecutive to the initial silver deposition with monovalent valence transfers. About 80% of H_2_ results from the reaction of the oxonium ion or the ion pair H_3_O^+^ F^−^ with the bulk silicon, according to Equation (2). The molecular hydrogen accounts for slightly less than 50% of the total H(+I) species reduction in the overall balance of the formation of H(0). The remaining part contributes to the formation of Si–H bonds (H_Si_).

Example 2, highlighted in green in Figure 2e,f is an arithmetic average of three experimental results in the range of an initial Ag^+^ molality between *b* Ag (*diss., t* = 0 s) = 1.3 × 10^−3^ and *b* Ag (*diss., t* = 0 s) = 1.9 × 10^−3^ mol∙kg^−1^. The participation of the oxonium ions in the silicon oxidation process is not possible at this level since the equilibrium potential of the 2H_3_O^+^/H_2_ half-cell is +0.05 V vs. SHE, which does not reach the energy level of the valence band of the bulk silicon at the silver/silicon contact of *E_V_* (Si_bulk_) = −0.38 eV (Figure 2a). However, the subscribed stoichiometric Δ*n* Ag*:*Δ*n* Si*:*Δ*n* H_2_ ratio of ≈3.06:1:0.37 mol:mol:mol indicates that Ag^+^ cannot be the only oxidant to oxidize Si to the oxidation state of +IV. Following the prominent theory of a mixed occurrence of divalent and tetravalent reactions according to Equations (3) and (4), H_2_O and/or HF or HF_2_^−^ would be plausible as secondary oxidants. Based on the simple valence balance regarding the redox pair Si/Si^4+^, four valences must be cumulatively contributed by the reduction of Ag^+^ to Ag and by the reduction of 2H(+I) to H_2_. Taking into account the analytical errors of the measurement methods, this balance would be approximately fulfilled. Considering the mean value of the Δ*n* Ag*:*Δ*n* Si*:*Δ*n* H_2_ ratio of ≈3.06:1:0.37 mol:mol:mol determined, slightly too little molecular hydrogen was formed or to little Ag^+^ was reduced to match the valence balance (3.06 × 1 h^+^ (Ag) + 0.37 × 2 h^+^ (H_2_) = 3.80 h^+^ < 4 h^+^).

The widespread explanation of the molecular hydrogen formation resulted from a divalent reaction process [17,21,34,36,37,38,39,40,41,42,43,44,45,46,47,48,49] in combination with the hydrogen-free reaction process during tetravalent valence transfer [11,12,15,21,34,40,41,44,55] may still fit, to some extent, based on the findings of example 2 of silver deposition. The results of the other metal depositions explained in Section 3.2, Section 3.3 and Section 3.4, especially the findings of platinum deposition, support a different and more complex theory for molecular hydrogen formation.

According to Figure 4, the molecular hydrogen formation is associated with an odd number of valence transfers from the metal-ion reduction to the Si–Si bonds of the bulk silicon, similar to the initial silver deposition reaction scheme shown in Figure 3a. What is different from the monovalent valence transfer shown in Figure 3a-1 is that the trivalent valence transfer (3 Ag^+^ → Ag + 3 h^+^) leads to a ≡Si–SiF_3_ intermediate and three dangling bonds (Figure 4(1a,2a)). Two of the dangling bonds combine to form a new Si–Si bond. The ≡Si–SiF_3_ intermediate is cleaved by water, favored by the destabilizing effect of the three adjacent Si–F bonds. Free hydrogen radical and the species SiF_3_OH are formed. Subsequently, the silanol group is replaced by F^−^ (Figure 4(2a,3a)). The dangling bond resulting from the cleavage of the ≡Si–SiF_3_ intermediate combines with the dangling bond still existing from the initial Si oxidation to form another Si–Si bond (Figure 4(3a,4a)). According to this reaction scheme, more molecular hydrogen would be produced than is indicated by the analytical findings of example 2 (Δ*n* Ag*:*Δ*n* Si*:*Δ*n* H_2_ = 3:1:0.5 mol:mol:mol). Consequently, there must be a second reaction process, which does not result in the formation of free hydrogen radicals and, thus, molecular hydrogen. A possible reaction scheme is sketched in Figure 4(1b–4b). It illustrates the reaction process for an even-numbered valence transfer from Ag^+^ to Si. In the case of a divalent or tetravalent valence transfer or the sum of both, as shown in Figure 4 (6 h^+^), there is no odd number of dangling bonds and no formation of ≡Si–SiF_3_ intermediate. Consequently, no subsequent reactions with H_2_O can proceed, which are suspected to be responsible for forming free hydrogen radicals and, consecutively, molecular hydrogen. Summarizing these considerations, about ≈12% of the reaction process was divalent (Δ*n* Ag*:*Δ*n* Si:Δ*n* H_2_ = 2:1:0 mol:mol:mol), ≈70% was trivalent (Δ*n* Ag:Δ*n* Si:Δ*n* H_2_ = 3:1:0.5 mol:mol:mol) and ≈18% was tetravalent (Δ*n* Ag:Δ*n* Si:Δ*n* H_2_ = 4:1:0 mol:mol:mol) to match the Δ*n* Ag*:*Δ*n* Si*:*Δ*n* H_2_ ratio of ≈3.06:1:0.37 mol:mol:mol determined.

Example 3, highlighted in blue in Figure 2e,f, is a summary of four experiments with an average initial Ag^+^ molality of *b* Ag (*diss., t* = 0 s) ≈ 1.2 × 10^−2^ mol∙kg^−1^ and a stoichiometric Δ*n* Ag:Δ*n* Si:Δ*n* H_2_ ratio of ≈3.57:1:0.34 mol:mol:mol. The average initial redox strength of the Ag^+^/Ag half-cells is *E* (Ag^+^/Ag) ≈ 0.66 V vs. SHE. Thus, the hydrogen-terminated silicon can be oxidatively attacked by the Ag^+^/Ag half-cell, following the energy plot in Figure 2a.

Similar to the reaction process with the bulk silicon, it is assumed that the formation of molecular hydrogen in the reaction of the Ag^+^/Ag half-cell with the hydrogen-terminated silicon is associated with the transfer of an odd number of holes. Similarly, the transfer of an even number of holes does not lead to the formation of molecular hydrogen. Figure 5 illustrates the reaction process proposed via a direct attack on the Si–H bonds inspired by the theories of Lehmann and Gösele [57] and Bertagna et al. [58]. According to this scheme, a divalent valence transfer to a Si–H bond leads to the formation of hydrogen ions and, in the subsequent reaction with water, to the formation of oxonium ions. A monovalent valence transfer to a Si–H bond results in the formation of free hydrogen radical and, consecutively, to molecular hydrogen (Figure 5(1,2)). The free binding sites on the oxidized silicon are occupied by F^−^. The resulting Si–F bonds destabilize the Si–Si back bonds. The consecutive reaction of water with the Si–Si back bonds produces hydrogen radicals, which bind with the Si atoms not oxidized and form new Si–H bonds, i.e., the new hydrogen-terminated silicon. On the side of the oxidized silicon, OH^−^ is bound and, subsequently, replaced by fluoride. The oxidized Si will be converted to the formal end product SiF_6_^2−^ in several analogous steps (Figure 5(3,4)).

The scheme outlined in Figure 5 indicates the reaction of 7 Ag^+^ with two =Si=H_2_ groups exemplarily, resulting in a stoichiometric Δ*n* Ag:Δ*n* Si:Δ*n* H_2_ ratio of 3.5:1:0.25 mol:mol:mol. Slightly more molecular hydrogen was formed in example 3, marked in Figure 2e,f. According to the reaction balance shown in Figure 5, it can be assumed that the reaction probably occurred more at the hydrogen-rich =Si=H_2_ and–Si≡H_3_ groups than at the hydrogen-poor ≡Si–H groups.

Example 4, highlighted in black in Figure 2e,f, summarizes two experiments with similar levels of initial Ag^+^ molality as example 3, but with a different mean stoichiometric Δ*n Ag:*Δ*n Si:*Δ*n H_2_* ratio of ≈4.28:1:0.02 mol:mol:mol. The reason for this finding can be found in the kinetically limited silicon dissolution in the two experiments of example 4. The anionic HF species F^−^, HF_2_^−^ and H_2_F_3_^−^ are responsible for the fast oxidized silicon dissolution process [26]. Their cumulative activity is less than six times the amount of silicon dissolution after *t* = 3600 s. This criterion was determined in our previous study and reflects the stoichiometric ratio of F and Si in the formal final product SiF_6_^2−^ [26]. If the activities of these species are below this criterion, the dissolution of the oxidized silicon slows down. This effect becomes stronger the further the critical threshold value is undershot. Consequently, in the series of experiments in this study, those with the lowest HF addition of 0.056 ± 0.003 mol∙kg^−1^ are particularly affected by this phenomenon, where example 4 belongs. The stoichiometric ratio deviating from example 3 can be explained as follows: If the surface ≡Si–H, =Si=H_2_ and–Si≡H_3_ groups are dissolved too slowly after the initial oxidation, further valences are transferred from the Ag^+^/Ag half-cell to these groups. This means that the hydrogen radicals formed by the initial oxidation of the Si–H bonds are then also oxidized, resulting in the formation of hydrogen ions and, consecutively, in the formation of oxonium ions. Consequently, the proportion of molecular hydrogen decreases with the increasing amount of valence transfers induced by the Ag^+^ reduction. If the hydrogen-rich =Si=H_2_ and–Si≡H_3_ groups tend to be attacked more than the hydrogen-poor ≡Si–H groups, stoichiometric Δ*n* Ag*:*Δ*n* Si ratios of > 4:1 mol:mol can result, since the hydrogen termination acts as a reducing agent in addition to the silicon. Comparing the reaction equations of examples 3 and 4 in Figure 5, it can be deduced that approximately the same proportions of Si–H_x_ groups were oxidized. However, due to the higher valence input by the Ag^+^/Ag half-cell per Si–H_x_ group, as in example 4, less molecular hydrogen and more oxonium ions were formed.

### 3.2. Copper Deposition onto Multicrystalline Silicon

In this series of experiments, 44 batch tests were performed with initial Cu^2+^ molalities between *b* Cu (*diss., t* = 0 s) = 1.2 × 10^−4^ and 1.5 × 10^−1^ mol∙kg^−1^ at three different levels of HF addition (0.060 ± 0.001 mol∙kg^−1^ HF, 0.305 ± 0.005 mol∙kg^−1^ HF, and 1.502 ± 0.024 mol∙kg^−1^).

Three redox half-cells must be considered to describe the copper deposition process on silicon. Based on the initial concentration range of Cu^2+^ and the activity coefficients calculated according to Bromley [79], the redox levels of the Cu^2+^/Cu half-cells [28] are between *E* (Cu^2+^/Cu) = 0.22 and 0.31 V vs. SHE. The redox level of the Cu^2+^/Cu^+^ half-cells is at most *E* (Cu^2+^/Cu^+^) = −0.08 to 0.09 V vs. SHE [28,79]. The Cu^+^/Cu half-cells can reach a maximum level of *E* (Cu^+^/Cu) = 0.28 to 0.47 V vs. SHE [28,79].

Given the energetic level of the Cu^2+^/Cu half-cells, the process of initial copper nucleation is only possible by a special state effect, given the higher initial valence band level of the bulk silicon and the hydrogen-terminated silicon (*E_V_* (Si_bulk_) = −0.41 eV and *E_V_* (Si-H_x_) = −0.68 eV, respectively, Figure 6a). Lim et al. [89] postulated that the initial copper deposition starts primarily at step edges, kinks and rough surfaces. The authors combined this observation with the thesis that the =Si=H_2_ groups are the key factor for the initial copper nucleation.

The same observations were made in this study. The growth of the copper nucleation started on the cut sides of the Si wafers and, subsequently, spread over the entire surface. According to the theory of Lim et al. [89], the valence exchange between Cu^2+^ and Si should have been light-induced via the conduction band of silicon because of the energetic level of the valence bands of bulk and hydrogen-terminated silicon are not accessible by all Cu^2+^/Cu^+^/Cu half-cells during the initial copper deposition. However, this consideration would not explain the selectivity of the attack on the kinks and edges. It is more probable that the primary copper deposition starts at exposed, non-hydrogen-terminated silicon atoms (Figure 6a). The standard potential of the Si/SiF_6_^2−^ half-cell is *E* (Si/SiF_6_^2−^) = −1.24 V vs. SHE [28], which is significantly lower than all Cu^2+^/Cu^+^/Cu half-cells [28]. From this point of view, the preferential deposition of copper at such points would be more plausible.

After the initial copper deposition, the Fermi energy equilibration at the copper/silicon contact leads to the bending of the conduction and valence bands of the silicon (Figure 6a). Since the differences in work function between copper and hydrogen-terminated silicon are larger than those between silver and hydrogen-terminated silicon [83,84,85,86,87,88], the amount of the band bending at the copper/silicon contact is greater. The upper limit of the valence band of the bulk silicon can be estimated from the threshold of oxonium-ion reduction at *b* Cu (*diss., t* = 0 s) = 2.5 × 10^−4^ mol∙kg^−1^ determined in our previous study [26]. The equilibrium potential of the 2H_3_O^+^/H_2_ half-cell at this Cu^2+^ molality is *Eq. Pot.* (2H_3_O^+^/H_2_) = +0.17 V vs. SHE (with *J* (Cu^2+^) = 1.7 × 10^−2^ mA∙cm^−2^) [26,33]. As a simple approximation, the energetic level of the valence band of the bulk silicon at the copper-silicon contact is *E_V_* (Si_bulk_) ≈ −0.17 eV. The upper limit of the valence band of the hydrogen-terminated silicon is obtained from the threshold at *b* Cu (*diss., t* = 0 s) = 8.2 × 10^−3^ mol∙kg^−1^. It was concluded in our previous study [26] that the ≡Si–H, =Si=H_2_ and–Si≡H_3_ groups are attacked by the Cu^+^/Cu half-cell above this level. The redox strength of the Cu^+^/Cu half-cell is *E* (Cu^+^/Cu) ≤ 0.38 V vs. SHE at this threshold. The upper limit of the valence band of the hydrogen-terminated silicon is correspondingly at most at *E* (Si-H_x_) ≈ −0.38 eV.

It can be concluded from the position of the valence bands of silicon at the copper/silicon contact that the Cu^2+^/Cu and Cu^+^/Cu half-cells and, up to the threshold of *b* Cu (*diss., t* = 0 s) = 2.5 × 10^−4^ mol∙kg^−1^, the 2H_3_O^+^/H_2_ half-cell can participate in the oxidation of silicon, but not the too weak Cu^2+^/Cu^+^ half-cell.

Figure 6b shows the half-cells’ redox levels considering the upper and lower limits of the initial Cu^2+^ molalities of the batch experiments and at the two threshold values mentioned above. It can be noted that Cu^2+^ was almost completely reduced to the metal state (Δ*n* Cu (*diss., t* = 3600 s)) in all experiments up to the threshold value of *b* Cu (*diss., t* = 0 s) = 8.2 × 10^−3^ mol∙kg^−1^. In the experiments with a higher Cu^2+^ concentration, Cu^2+^ was not completely reduced and deposited. There is a plateau of Δ*n* Cu (*diss., t* = 3600 s) in each case as a function of initial HF molality. A maximum amount of (5.2 ± 0.5) × 10^−4^ mol Cu was deposited in the experiments with an HF molality of 0.060 ± 0.001 mol∙kg^−1^. The maximum amounts within the batches with a higher HF molality were (1.3 ± 0.5) × 10^−3^ mol Cu (0.305 ± 0.005 mol∙kg^−1^ HF) and (4.2 ± 0.9) × 10^−3^ mol Cu (1.502 ± 0.024 mol∙kg^−1^ HF), respectively.

Similarly, the analytical data in Figure 6c show an increase in the amount of silicon dissolution (Δ*n* Si (*diss., t* = 3600 s)) in correlation with the initial Cu^2+^ molality until the threshold of *b* Cu (*diss., t* = 0 s) = 8.2 × 10^−3^ mol∙kg^−1^ is reached. The range of Δ*n* Si (*diss., t* = 3600 s) values varies from 8.9 × 10^−6^ mol at *b* Cu (*diss., t* = 0 s) = 1.2 × 10^−4^ mol∙kg^−1^ to Δ*n* Si *(diss., t* = 3600 s) = 3.1 × 10^−4^ mol at the threshold. Above the threshold, the amount of silicon dissolution decreases with increasing Cu^2+^ molality for the batch experiments with the lowest HF molality (0.060 ± 0.001 mol∙kg^−1^ HF). At *b* Cu (*diss., t* = 0 s) = 6.2 × 10^−2^ mol∙kg^−1^, Δ*n* Si (*diss., t* = 3600 s) is about 1.5 × 10^−4^ mol. The silicon dissolution in the solutions with 0.305 ± 0.005 mol∙kg^−1^ HF increases to a maximum value of Δ*n* Si (*diss., t* = 3600 s) = 9.2 × 10^−4^ mol up to *b* Cu (*diss., t* = 0 s) = 3.4 × 10^−2^ mol∙kg^−1^ and then decreases to Δ*n* Si (*diss., t* = 3600 s) = 4.6 × 10^−4^ mol at *b* Cu (*diss., t* = 0 s) = 7.7 × 10^−2^ mol∙kg^−1^. The silicon dissolution in the HF richest solutions (1.502 ± 0.024 mol∙kg^−1^) increases to *b* Cu (*diss., t* = 0 s) = 7.8 × 10^−2^ mol∙kg^−1^ and remains at the level of Δ*n* Si (*diss., t* = 3600 s) = (2.3 ± 0.5) × 10^−3^ mol.

The limitations of Δ*n* Si (*diss., t* = 3600 s) correspond to the product of the cumulative activity of the anionic HF dissociation species (*a* F^−^, HF_2_^−^, H_2_F_3_^−^) [79,90], the mass of the respective solution (*m_Sol_*) and the factor 0.5. This means that at least two anionic HF species are required to dissolve one oxidized silicon species. Otherwise, the oxidized silicon dissolution and, subsequently, the further copper deposition will be stopped.

The results of the H_2_ measurements are shown in Figure 6d. The formation of molecular hydrogen occurred in all 44 individual experiments. Referenced to the initial metal ion molality, the stoichiometric amounts of the molecular hydrogen formation for silver and copper deposition are at approximately the same level. Analogous to the silver deposition findings, the lowest amount of the molecular hydrogen formation was detected in the experiment with the lowest initial Cu^2+^ molality with Δ*n* H_2_ (*g, t* = 3600 s) ≈ 2.4 × 10^−6^ mol. Up to the threshold value of *b* Cu (*diss., t* = 0 s) = 8.2 × 10^−3^ mol∙kg^−1^, the molecular hydrogen formation increases to about Δ*n* H_2_ (*g, t* = 3600 s) ≈ 4.9 × 10^−5^ mol, regardless of the set HF molality. The molecular hydrogen formation for the solutions with the HF molality of 0.060 ± 0.001 mol∙kg^−1^ decreased continuously above the threshold with increasing initial Cu^2+^ molality and was Δ*n* H_2_ (*g, t* = 3600 s) ≈ 1.8 × 10^−5^ mol at *b* Cu (*diss., t* = 0 s) = 6.2 × 10^−2^ mol∙kg^−1^. The molecular hydrogen formation reached its maximum of Δ*n* H_2_ (*g, t* = 3600 s) ≈ 9.9 × 10^−5^ mol at *b* Cu (*diss., t* = 0 s) = 2.2 × 10^−2^ mol∙kg^−1^ for the solutions with an initial HF molality of 0.305 ± 0.005 mol∙kg^−1^. It decreased continuously with the increasing initial Cu^2+^ molality until Δ*n* H_2_ (*g, t =* 3600 s) ≈ 2.9 × 10^−5^ mol at *b* Cu (*diss., t* = 0 s) = 7.7 × 10^−2^ mol∙kg^−1^. In the case of the HF richest solutions (1.502 ± 0.024 mol∙kg^−1^ HF), the maximum molecular hydrogen formation was Δ*n* H_2_ (*g, t* = 3600 s) ≈ 3.3 × 10^−4^ mol measured at *b* Cu (*diss., t* = 0 s) = 9.0 × 10^−2^ mol∙kg^−1^. The amounts of molecular hydrogen formation above this Cu^2+^ level decreased to a plateau of Δ*n* H_2_ (*g, t* = 3600 s) = (1.4 ± 0.1) × 10^−4^ mol. The amount of molecular hydrogen formation for the experiment with the highest initial Cu^2+^ molality with *b* Cu (*diss., t* = 0 s) = 5.0 × 10^−1^ mol∙kg^−1^ was Δ*n* H_2_ (*g, t* = 3600 s) ≈ 1.0 × 10^−4^ mol.

Figure 6e shows the copper deposition and silicon dissolution findings concerning each other. Analogous to the silver deposition, the smallest stoichiometric Δ*n* Cu*:*Δ*n* Si *(diss., t* = 3600 s) ratio of 0.72:1 mol:mol is detected for the solution with the lowest initial Cu^2+^ molality of *b* Cu (*diss., t* = 0 s) = 1.2 × 10^−4^ mol∙kg^−1^. As the initial Cu^2+^ molality increases, the Δ*n* Cu*:*Δ*n* Si *(diss., t* = 3600 s) ratios also increase sigmoidally. Above the threshold of *b* Cu (*diss., t* = 0 s) = 8.2 × 10^−3^ mol∙kg^−1^, the stoichiometric Δ*n* Cu*:*Δ*n* Si (*diss., t* = 3600 s) ratios range between 1.5:1 mol:mol and 3:1 mol:mol after *t* = 3600 s reaction time. According to the findings of our previous study [26], the Δ*n* Cu*:*Δ*n* Si (*diss., t* = 3600 s) ratios can change in a range of 3:1 and 1:2 mol:mol under these process conditions. The Δ*n* Cu*:*Δ*n* Si (*diss., t* = 3600 s) ratios over the entire Cu^2+^ concentration range vary according to the minima and maxima functions plotted [26]. Therefore, the findings shown in Figure 6e are to be understood as time-weighted averages of varying Δ*n* Cu*:*Δ*n* Si *(diss., t* = 3600 s) ratios.

Figure 6f shows the stoichiometric ratios of the molecular hydrogen formation and the silicon dissolution. Similar to the silver deposition, the Δ*n* H_2_*:*Δ*n* Si (*g* or *diss., t* = 3600 s) ratios decrease sigmoidally with increasing initial Cu^2+^ molality. The Δ*n* H_2_*:*Δ*n* Si (*g* or *diss., t* = 3600 s) ratios are largest ((0.44 ± 0.05):1 mol:mol to (0.42 ± 0.05):1 mol:mol) in the range of the oxonium-ion reduction process enabled (red indicated). The ratios decrease beyond the threshold of *b* Cu (*diss., t* = 0 s) = 2.5 × 10^−4^ mol∙kg^−1^ to a level of Δ*n* H_2_*:*Δ*n* Si (*g* or *diss., t* = 3600 s) = (0.16 ± 0.05):1 mol:mol up to the threshold of *b* Cu (*diss., t* = 0 s) = 8.2 × 10^−3^ mol∙kg^−1^. At the maximum of the chosen initial Cu^2+^ molality of *b* Cu (*diss., t* = 0 s) = 1.0 × 10^−2^ mol∙kg^−1^, the Δ*n* H_2_*:*Δ*n* Si (*g* or *diss., t =* 3600 s) ratio is about (0.08 ± 0.03):1 mol:mol.

Three examples of the stoichiometric ratios of copper deposition, silicon dissolution and molecular hydrogen formation (Δ*n* Cu*:*Δ*n* Si*:*Δ*n* H_2_) were extracted from Figure 6e,f. Based on the ratios determined, the possible reaction processes were discussed in view of the findings of the energy scheme visualized in Figure 6a.

Example 1, highlighted in red in Figure 6e,f, is a single experiment at *b* Cu (*diss., t* = 0 s) = 1.5 × 10^−4^ mol∙kg^−1^. The stoichiometric Δ*n* Cu*:*Δ*n* Si*:*Δ*n* H_2_ ratio was determined to ≈0.94:1:0.46 mol:mol:mol. Figure 7a-1,2 briefly outlines the process of the initial copper deposition. Cu^2+^ is probably reduced directly to Cu during the initial metal nucleation. The reaction is expected to skip the formation of Cu^+^ because the Cu^2+^/Cu^+^ half-cell is too weak to reach the valence band of a silicon atom. According to the reaction scheme in Figure 7a-1, two Cu^2+^ ions are reduced on the cathodic side, and, cumulatively, four electronic holes are transferred to one Si atom in the bulk silicon. Formally four dangling bonds are formed on the surrounding four Si atoms not oxidized, which connect consecutively to two Si–Si bonds without leaving an unpaired electron. Consequently, there is no consecutive water reaction and, thus, no free hydrogen radicals and no molecular hydrogen formation. Subsequently, four fluoride ions attach to the Si^4+^ species, and one or more species containing fluoride converts the intermediate SiF_4_ species to the formal final product SiF_6_^2−^ (Figure 7a-2). According to this reaction scheme, the molecular hydrogen detected in the experiment must have been formed predominantly via the oxonium-ion reduction that starts after the initial copper deposition. The reaction process is shown in a shortened form in Figure 7a-3,4. According to this scheme, the reduction of the oxonium ion or the H_3_O^+^ F^−^ ion pair, proposed in Equation (2), creates one hole, which is transferred to a Si–Si bond, generating one free hydrogen radical and water. On the anodic side, a dangling bond and a Si–F bond are formed. The water, resulting from the oxonium-ion reduction, reacts consecutively with an adjacent destabilized Si–Si bond of the pre-oxidized Si^+^ species to form another free hydrogen radical and another dangling bond. Both hydrogen radicals combine to form one hydrogen molecule. The two dangling bonds form a new Si–Si bond. A second Si–F bond forms on the Si^2+^ species. The two water molecules subsequently react with the remaining two destabilized Si^2+^–Si bonds. One hydrogen radical and one hydroxide group are formed per broken Si–Si bond. The hydrogen radicals bind to the silicon atoms not oxidized and OH^−^ to the Si^4+^ species. Subsequently, the intermediate Si–OH groups are replaced by Si–F bonds and, finally, the formal end product SiF_6_^2−^ is formed. In the sum of the processes of copper deposition and oxonium-ion reduction, a Δ*n* Cu*:*Δ*n* Si*:*Δ*n* H_2_ ratio of 1:1:0.5 mol:mol:mol results, according to the scheme outlined in Figure 7a, which corresponds approximately to the ratio of the analytical findings. The overall reaction equation formulated according to the analytical finding is shown below the reaction scheme.

Example 2, highlighted in green in Figure 6e,f, is an averaged result of four individual experiments at an initial average Cu^2+^ molality of *b* Cu (*diss., t* = 0 s) ≈ 2.2 × 10^−3^ mol∙kg^−1^. The average stoichiometric Δ*n* Cu*:*Δ*n* Si*:*Δ*n* H_2_ ratio is ≈1.54:1:0.29 mol:mol:mol. The Cu^2+^/Cu half-cell has a redox strength of *E* (Cu^2+^/Cu) ≈ 0.26 V vs. SHE. At this level, it can interact with the bulk silicon (Figure 6a). The 2H_3_O^+^/H_2_ half-cell, with an equilibrium potential of *Eq. Pot.* (2H_3_O^+^/H_2_) ≈ 0.00 V vs. SHE, and the Cu^2+^/Cu^+^ half-cell with a redox level of maximum *E* (Cu^2+^/Cu^+^) ≈ −0.01 V vs. SHE, cannot be involved in the oxidation of silicon. Based on this consideration, only the divalent and tetravalent reaction mechanism can occur between the Cu^2+^/Cu half-cell and the bulk silicon. According to this study’s theory, both processes go ahead without the formation of molecular hydrogen when reacting with the Si–Si bonds (Figure 7b-1–3). Consequently, the molecular hydrogen formation observed must originate from the Si–H bonds. However, the Si–H bond cannot be directly oxidized by the Cu^2+^/Cu half-cells due to the energy level of the hydrogen-terminated silicon, shown in Figure 6a. Nevertheless, the oxidative attack on the Si–Si back bonds of the hydrogen-terminated silicon is plausible. As a result, the intermediate HSiF_3_ postulated by Gerischer et al. [61], Kooij and Vanmaekelbergh [62], Kolasinski [63], and Stumper and Peter [36] can be formed. This species would be further converted to SiF_6_^2−^ and molecular hydrogen according to the scheme of Equation (6) [63] or as a result of a reaction with water and F^−^, as outlined in Figure 7b-4. According to that scheme, an overall stoichiometric Δ*n* Cu*:*Δ*n* Si*:*Δ*n* H_2_ ratio of ≈ 1.5:1:0.17 mol:mol:mol would result. Since about 1.7 times the molecular hydrogen was formed according to the analytical findings of Example 2, it can be concluded that not only the Si–Si back bonds of the ≡Si–H groups but instead the Si–Si back bonds of the more hydrogen-rich =Si=H_2_ and possibly also those of the–Si≡H_3_ groups are attackable. This was considered in the calculation of the overall reaction in Figure 7b.

Example 3, highlighted in blue in Figure 6e,f, is the average of two experiments with an averaged Cu^2+^ molality of *b* Cu(*diss., t* = 0 s) ≈ 2.6 × 10^−1^ mol∙kg^−1^ and a Δ*n* Cu*:*Δ*n* Si*:*Δ*n* H_2_ ratio of about 2.21:1:0.08 mol:mol:mol. The redox level of the Cu^2+^/Cu half-cells is *E* (Cu^2+^/Cu) = 0.30 V vs. SHE. If the Δ*n* Cu*:*Δ*n* Si *(diss., t* = 3600 s) ratio exceeds an amount of 2:1 mol:mol, more than four valences must have been transferred to the silicon. Consequently, Si cannot be the only reducing agent. It is concluded that the Si–H bonds were attacked, and the hydrogen of the surface ≡Si–H, =Si=H_2_ and –Si≡H_3_ groups has acted as a second reducing agent. According to the energy diagram of Figure 6a, the oxidation may not have occurred through the Cu^2+^/Cu half-cell but through the Cu^+^/Cu half-cell with a redox strength of *E* (Cu^+^/Cu) = 0.46 V vs. SHE. The formation of Cu^+^ as an intermediate in the reduction of Cu^2+^ to Cu is, in fact, unlikely, since the Cu^2+^/Cu^+^ half-cell with a redox strength of, at most, *E* (Cu^2+^/Cu^+^) = 0.09 V vs. SHE would not have been strong enough for a reaction with either the bulk or hydrogen-terminated silicon. Alternatively, the formation of Cu^+^ may occur by a comproportionation reaction of the Cu^2+^ in solution with the metallic copper at its interface. This reaction is normally not energetically favored; instead, it is the disproportionation of Cu^+^. To enable such a comproportionation, the Cu^2+^ reduction in interaction with Si would have been significantly inhibited. As has already been described, for example, 4 of the silver deposition, the limitation of the silicon dissolution is possible due to insufficient activity of the F^−^, HF_2_^−^ and H_2_F_3_^−^ species responsible for the fast silicon dissolution [26]. This is also the situation, for example, 3 of the copper deposition. In this case, the ratio between the cumulative activity of F^−^, HF_2_^−^ and H_2_F_3_^−^ [79,90] and the molality of the dissolved Si is about 1:1 mol∙kg^−1^. In view of this, an oxidative attack on the bulk silicon first occurs through the Cu^2+^/Cu half-cell, as sketched in Figure 7c. Since the oxidized silicon is not dissolved fast enough, Cu^2+^ undergoes a very slow [26] comproportionation reaction with the Cu deposited previously to form Cu^+^. The Cu^+^ produced is strong enough to oxidize the Si–H groups.

Since monovalent valence transfers occur under this condition, free hydrogen radicals and, subsequently, molecular hydrogen can be generated in the reaction with the ≡Si–H, =Si=H_2_ and–Si≡H_3_ groups. However, according to the analytical findings on the low molecular hydrogen formation, the number of cumulative double monovalent valence transitions must have been in the majority so that hydrogen ions and, subsequently, oxonium ions were formed predominantly. Based on the overall reaction balance, more hydrogen-rich =Si=H_2_ and–Si≡H_3_ groups must have been attacked, possibly due to the slow dissolution of the former Si–H groups, which had already been oxidized in advance (Figure 7c).

### 3.3. Gold Deposition onto Multicrystalline Silicon

AuCl_4_^−^ was used as the oxidant within ten gold deposition experiments. The initial molality of this species ranged between *b* Au (*diss., t* = 0 s) = 4.8 × 10^−7^ and 7.6 × 10^−2^ mol∙kg^−1^ at an HF level of 1.333 ± 0.010 mol∙kg^−1^, resulting in a range of redox strength of the AuCl_4_^−^/Au half-cell [28] from *E* (AuCl_4_^−^/Au) = 0.88 to 0.97 V vs. SHE. The reduction of AuCl_4_^−^ to silicon theoretically also involves other half-cells [21]. Furthermore, pH-dependent ligand exchange changes the complex structure of AuCl_4_^−^ [91]. Both aspects remain disregarded in the formal consideration of the reaction processes based on the stoichiometric findings.

The energy diagram sketched in Figure 8a shows that the AuCl_4_^−^/Au half-cell was sufficiently strong to reach the valence band of the bulk silicon and that of the hydrogen-terminated silicon in each of the experiments. Due to the larger difference in the work function between gold and hydrogen-terminated silicon compared to the findings of the silver and copper deposition [83,84,85,86,87,88], a much stronger conduction and valence-band bending of the silicon at the gold/silicon contact results. This influences the valence transfer between AuCl_4_^−^ and silicon. In our previous study [26], two thresholds were identified. The first is at *b* Au (*diss., t* = 0 s) = 3.1 × 10^−5^ mol∙kg^−1^. The kinetics of gold deposition is maximal at this concentration level. Above this value, the kinetics decreases and reaches a local minimum at the second threshold at *b* Au (*diss., t* = 0 s) = 1.5 × 10^−4^ mol∙kg^−1^.

These findings can be interpreted as follows: The valence bands of silicon at the first threshold have been bent by +0.41 eV so that the valence band of bulk silicon is in the range of the Fermi energy (*E_V_* (Si_bulk_) ≈ *E_F_* = 0 eV) and the valence band of hydrogen-terminated silicon is at *E_V_* (SiH_x_) = −0.27 eV (= −0.68 eV + 0.41 eV). Assuming that the holes are transferred from the metal to the silicon at the Fermi energy level, the kinetics of the hole transfer is maximum under these circumstances [26].

If the work function of the gold exceeds that of the hydrogen-terminated silicon by more than the amount of +0.41 eV, the valence band of the bulk silicon is bent above the level of the Fermi energy and a potential barrier, the Schottky barrier, is created (*E_V_* (Si_bulk_) > 0 eV). This inhibits the valence transfer to the bulk silicon, and the valence transfer occurs preferentially via the valence band of the hydrogen-terminated silicon because its energy level is still below the Fermi energy (*E_V_* (Si-H_x_) < 0 eV). An effective change in the work function of the gold is possible when the concentration of negative charges at the interface between the gold surface and the HF solution increases due to an increasing amount of adsorbed AuCl_4_^−^, in correlation with a higher AuCl_4_^−^ molality in the solution. The resulting rising dipole moment above the gold surface increases the energy required for valence transfer from the gold surface. Such a relationship has been proven by Gossenberger et al. [92] for halides on a platinum/silicon contact. Furthermore, there is also a dependence of the work function on metal and silicon’s lattice orientation [83,84,85,86,87,88]. However, its importance cannot be estimated by the use of multicrystalline silicon in our previous study [26] and this study.

The second threshold at *b* Au (*diss., t* = 0 s) = 1.5 × 10^−4^ mol∙kg^−1^ is the upper limit of the oxonium-ion reduction process. The 2H_3_O^+^/H_2_ equilibrium potential *Eq. Pot.* (2H_3_O^+^/H_2_) is 0.15 V vs. SHE [33] based on the current density of *J* (Au^3+^) = 1.6 × 10^−2^ mA∙cm^−2^ determined [26] at this level. At the gold/silicon contact, this amount is equivalent to the energetic upper limit of the valence band of hydrogen-terminated silicon (*E_V_* (Si-H_x_) = −0.15 eV).

The energetic level of the valence band of the bulk silicon should be at *E_V_* (Si_bulk_) = +0.12 eV (= −0.41 eV + 0.53 eV), with an equal amount of band bending of +0.53 eV as the valence band of the hydrogen-terminated silicon (= −0.15—−0.68 eV). The resulting potential barrier no longer allows the valence transfer to the bulk silicon. The valence transfer occurs only to the hydrogen-terminated silicon. The Schottky barrier can also increase even greater with increasing AuCl_4_^−^ molality, as outlined in Figure 8a. However, no further threshold was identified in the kinetics of gold deposition in our previous study [26] until *b* Au (*diss., t* = 0 s) = 9.3 × 10^−2^ mol∙kg^−1^. Thus, there is no indication of band bending of the hydrogen-terminated silicon at the gold/silicon contact above the level of the Fermi energy.

Figure 8b plots the stoichiometric amounts of the gold deposition within the period of *t =* 3600 s (Δ*n* Au (*diss., t* = 3600 s)) relative to the initial AuCl_4_^−^ molality *b* Au (*diss., t* = 0 s). In addition, the redox levels of AuCl_4_^−^ and 2H_3_O^+^/H_2_ half-cells at the upper and lower limits of the initial AuCl_4_^−^ molalities and the thresholds of *b* Au (*diss., t* = 0 s) = 3.1 × 10^−5^ mol∙kg^−1^ and 1.5 × 10^−4^ mol∙kg^−1^ are indexed.

The results of the experiments in which the oxonium ions participated in addition to the AuCl_4_^−^ ions are marked in red. The black rectangles symbolize the results due to the oxidation of the hydrogen-terminated silicon by the AuCl_4_^−^ half-cells. The comparison of the product of the initial AuCl_4_^−^ molality *b* Au (*diss., t* = 0 s) and the mass of solution *m_sol_* with the amount of gold deposition Δ*n* Au (*diss., t* = 3600 s) reveals that AuCl_4_^−^ was not completely reduced to Au within *t =* 3600 s (≈ 43 to 77%) in the two experiments with the lowest initial AuCl_4_^−^ molalities. For the other experiments, the AuCl_4_^−^ reduction to Au occurred over 99.7% during this period.

The amount of silicon dissolution (Figure 8c) has its minimum in the experiment with the lowest initial AuCl_4_^−^ molality of *b* Au (*diss., t* = 0 s) = 2.9 × 10^−8^ mol∙kg^−1^ with Δ*n* Si *(diss., t* = 0 s) = 2.6 × 10^−6^ mol. The amounts of Δ*n* Si (*diss., t* = 3600 s) increase mathematically in a potential function to Δ*n* Si (*diss., t* = 3600 s) ≈ 1.2 × 10^−5^ mol up to the threshold of the oxonium-ion reduction at *b* Au (*diss., t* = 0 s) = 1.5 × 10^−4^ mol∙kg^−1^. Above this threshold, the slope of silicon dissolution regarding the initial AuCl_4_^−^ molality is significantly larger. The silicon dissolution Δ*n* Si (*diss., t* = 3600 s) is about 4.4 × 10^−3^ mol at the maximum initial AuCl_4_^−^ molality of *b* Au (*diss., t* = 0 s) = 7.6 × 10^−2^ mol∙kg^−1^.

The formation of molecular hydrogen in gold deposition (Figure 8d) occurs on a similar magnitude to that in silver and copper deposition up to the limit of the oxonium-ion reduction. The amount of Δ*n* H_2_ (*g, t =* 3600 s) in the two experiments with the lowest initial AuCl_4_^−^ molality is between ≈1.8 × 10^−6^ and ≈2.0 × 10^−6^ mol, just above the analytical detection limit. At the threshold of the oxonium-ion reduction, Δ*n* H_2_ (*g, t* = 3600 s) is extrapolated to be ≈5.8 × 10^−6^ mol. Above this level, the amount of the molecular hydrogen formation increases much more sharply relative to that of silver and copper deposition in relation to the initial metal ion molality, reaching a value of Δ*n* H_2_ (*g, t* = 3600 s) = 1.7 × 10^−3^ mol at *b* Au (*diss., t* = 0 s) = 7.6 × 10^−2^ mol∙kg^−1^. This value is the absolute maximum value of all experiments in this study.

Figure 8e,f shows the stoichiometric ratios of gold deposition and silicon dissolution (Δ*n* Au*:*Δ*n* Si (*diss., t* = 3600 s)) and the formation of molecular hydrogen vs. the silicon dissolution (Δ*n* H_2_*:*Δ*n* Si *(g* or *diss., t* = 3600 s)), each in relation to the initial AuCl_4_^−^ molality. Analogously to the findings for silver and copper deposition, the Δ*n* Au*:*Δ*n* Si *(diss., t* = 3600 s) ratios are small at the initial AuCl_4_^−^ molalities below the oxonium-ion reduction threshold. Between *b* Au (*diss., t* = 0 s) = 4.8 × 10^−7^ and 7.6 × 10^−6^ mol∙kg^−1^, the Δ*n* Au*:*Δ*n* Si (*diss., t* = 3600 s) ratio is about 0.02:1 mol:mol. The Δ*n* Au*:*Δ*n* Si *(diss., t* = 3600 s) ratio grows sigmoidally with increasing initial AuCl_4_^−^ molality and reaches a level of ≈1:1 mol:mol at *b* Au (*diss., t* = 0 s) = 1.0 × 10^−3^ mol∙kg^−1^. The findings of the Δ*n* Au*:*Δ*n* Si *(diss., t* = 3600 s) ratios at higher initial AuCl_4_^−^ molalities vary between ≈1:1 and ≈1.1:1 mol:mol. As with the other metal depositions, these results should be understood as time-weighted stoichiometric Δ*n* Au*:*Δ*n* Si (*diss., t* = 3600 s) ratios in the period between *t* = 0 s and *t* = 3600 s. The variation of Δ*n* Au*:*Δ*n* Si (*diss., t* = 3600 s) ratios during the deposition process was elucidated in our previous study. The result is shown in Figure 8e as sigmoidal minima and maxima functions [26]. The Δ*n* H_2_*:*Δ*n* Si (*g* or *diss., t* = 3600 s) ratios behave inversely sigmoidally in relation to the initial AuCl_4_^−^ molality (Figure 8f). The Δ*n* H_2_*:*Δ*n* Si *(g* or *diss., t* = 3600 s) ratios are largest (between 0.59 ± 0.21:1 mol:mol and 0.58 ± 0.21:1 mol:mol) in the range of oxonium-ion reduction enabled between *b* Au (*diss., t* = 0 s) = 4.8 × 10^−7^ and 1.5 × 10^−4^ mol∙kg^−1^. Beyond that level, the Δ*n* H_2_*:*Δ*n* Si (*g* or *diss., t* = 3600 s) ratios decrease to a local minimum of 0.27:1 mol:mol at *b* Au (*diss., t* = 0 s) = 1.0 × 10^−3^ mol∙kg^−1^ and then increase again with increasing initial AuCl_4_^−^ molality to a stoichiometric ratio of Δ*n* H_2_:Δ*n* Si *(g* or *diss., t* = 3600 s*)* ≈ 0.39:1 mol:mol at *b* Au (*diss., t* = 0 s) = 7.2 × 10^−2^ mol∙kg^−1^.

Three examples were selected from the Δ*n* Au*:*Δ*n* Si*:*Δ*n* H_2_ ratios shown in Figure 8e,f to describe the reaction processes.

Example 1, highlighted in red in Figure 8e,f, summarizes the two experiments with the lowest initial AuCl_4_^−^ molalities (*b* Au (*diss., t* = 0 s) = 4.8 × 10^−7^ and 7.6 × 10^−6^ mol∙kg^−1^). The stoichiometric Δ*n* Au*:*Δ*n* Si*:*Δ*n* H_2_ ratio averages to ≈0.02:1:0.59 mol:mol:mol. The processes of the initial gold deposition and the consecutive oxonium-ion reduction are responsible for this finding. It is visualized in a short form in Figure 9a. The initial gold deposition is associated with a trivalent valence transfer from AuCl_4_^−^ to the bulk or hydrogen-terminated silicon (AuCl_4_^−^ → Au + 4 Cl^−^ + 3 h^+^). In both cases, the valence transfer is accompanied by the formation of one hydrogen radical (Figure 9a-1a,2a). In the reaction with the bulk silicon, the hydrogen radical is formed in the consecutive reaction of water with the intermediate ≡Si–SiF_3_ that occurs after the initial AuCl_4_^−^ reduction and Si–Si bond breakage, analogous to the illustration of the reaction sequence for silver deposition in Figure 4(1a–4a). In the process of three holes transfer to one =Si=H_2_ group, the molecular hydrogen formation would come from the one-sided bond break of the Si–H bond with the intermediate free hydrogen radical production. In each case, the Δ*n* Au*:*Δ*n* Si*:*Δ*n* H_2_ ratio would be 1:1:0.5 mol:mol:mol. Alternatively, a cumulative even-numbered valence transfer to the bulk silicon and the hydrogen-terminated silicon would also be possible, according to the reaction scheme of Figure 9a-1b,2b, without molecular hydrogen formation. A mixture of the two processes is conceivable since the stoichiometric Δ*n* H_2_*:*Δ*n* Si *(g* or *diss., t* = 3600 s) ratio near the threshold of oxonium-ion reduction at *b* Au *(diss., t =* 0 s) = 1.5 × 10^−4^ mol∙kg^−1^ is slightly less than 0.5:1 mol:mol (Figure 8f and Figure 9a(1,2)), but the processes involving the molecular hydrogen formation seem to be dominant. For the overall balance, however, the process of gold deposition on silicon is almost insignificant for the example 1 selected. According to the valence balance based on the stoichiometric Δ*n* Au*:Δn* Si (*diss., t* = 3600 s) ratio of 0.02:1 mol:mol, only 0.06 valences on average are transferred from the AuCl_4_^−^ reduction to one silicon atom. The remaining 3.94 valences per silicon must be attributed to the oxonium ion or H_3_O^+^ F^−^ and its intermediate product H_2_O.

The oxidative attack on the bulk silicon and hydrogen-terminated silicon in the oxonium-ion reduction process at the gold/silicon contact is possible due to the strong band bending of the valence bands of the silicon. In the case of an oxidative attack on a ≡Si–H group, according to the scheme in Figure 9a-3a,4a, one free hydrogen radical is produced from the reduction of the oxonium ion on the Si–H bond (Equation (2)). The second free hydrogen radical comes from the Si–H bond itself. Both free hydrogen radicals combine to form one hydrogen molecule. As a result, a stoichiometric Δ*n* H_2_*:*Δ*n* Si ratio of 1:1 mol:mol is obtained. An analogous attack on the more hydrogen-rich =Si=H_2_ and–Si=H_3_ groups would result in stoichiometric Δ*n* H_2_*:*Δ*n* Si ratios of 2:1 and 3:1 mol:mol, respectively. However, then the stoichiometric Δ*n* H_2_*:*Δ*n* Si (*g* or *diss., t* = 3600 s) ratio determined of 0.59:1 mol:mol, this can be only a marginal phenomenon.

According to the reaction scheme in Figure 9a-3b,4b, the oxidative attack on the bulk silicon results in free hydrogen radical due to the oxonium-ion reduction (Equation (2)) and the bond break of one Si–Si bond with dangling bond formation. The consecutive reaction of the water produced with the dangling bond forms another hydrogen radical. Both hydrogen radicals combine to form one hydrogen molecule. Furthermore, two Si^+^ intermediates are formed, which are converted to 2 SiF_6_^2−^ by consecutive hydrogen gas-free reaction with water and fluoride. Overall, this process results in a Δ*n* H_2_*:*Δ*n* Si ratio of 0.5:1 mol:mol. As a consequence of the average analytical finding of Δ*n* H_2_*:*Δ*n* Si *(g* or *diss., t* = 3600 s) ratio ≈ 0.59:1 mol:mol, the oxonium-ion reduction should have caused the oxidation of the bulk silicon predominantly (about 80%) and, to a lesser extent, the oxidation of the hydrogen-terminated silicon (about 20%). The weighting of these processes is reflected in the reaction balance in Figure 9a.

In Example 2, highlighted in blue in Figure 8e,f, the initial AuCl_4_^−^ molality is about *b* Au (*diss., t* = 0 s) = 1.0 × 10^−3^ mol∙kg^−1^. The oxidative attack in this range occurs exclusively by the AuCl_4_^−^/Au half-cell on the hydrogen-terminated silicon without coexisting oxonium-ion reduction. The stoichiometric Δ*n* Au*:*Δ*n* Si*:*Δ*n* H_2_ ratio is ≈1:1:0.27 mol:mol:mol. According to the reaction scheme outlined in Figure 9b, the formation of the molecular hydrogen is a consequence of many odd-numbered valence transfers from the AuCl_4_^−^ reduction to the Si–H bonds of the ≡Si–H, =Si=H_2_ and–Si≡H_3_ groups. The Si–Si back bonds of the hydrogen-terminated silicon are probably cleaved consecutively by water after the formation of Si–F bonds instead of the initially broken Si–H bonds. Hydroxyl groups are attached to the oxidized Si species and hydrogen radicals to the underlying Si atoms. The balance equation in Figure 9b, based on the stoichiometric Δ*n* Au*:*Δ*n* Si:Δ*n* H_2_ ratio determined, indicates that the ≡Si–H and =Si=H_2_ groups must have been attacked 40% each and the–Si≡H_3_ group about 20%.

The reaction steps in example 3 (marked in black in Figure 8e,f), with an initial AuCl_4_^−^ molality of *b* Au (*diss., t* = 0 s) = 7.2 × 10^−2^ mol∙kg^−1^ are basically the same as in example 2 (Figure 9b). The difference is the higher amount of molecular hydrogen formation in example 3. The Δ*n* Au*:*Δ*n* Si:Δ*n* H_2_ ratio is ≈1.03:1:0.39 mol:mol:mol. The background of this finding can be understood by comparing the balances of the reactions of examples 2 and 3 in Figure 9b. Apparently, the proportion of oxidized ≡Si–H, =Si=H_2_ and –Si≡H_3_ groups shifts from the hydrogen-poor ≡Si–H groups toward the more hydrogen-rich =Si=H_2_ and –Si≡H_3_ groups with increasing initial AuCl_4_^−^ molality. This leads to the conclusion that the more that silicon is surrounded by hydrogen bonds, the more difficult it is to attack. This can be justified by the higher binding energy of Si–H bonds (3.015 eV) compared to the Si–Si bonds (1.791 eV) [93]. Furthermore, Papaconstantopoulos and Economou have shown computationally that the energetic position of the valence band of silicon, but not that of the conduction band, is depressed in addition to the level of the Fermi energy with an increasing amount of hydrogen at the silicon [94]. Consequently, the AuCl_4_^−^/Au half-cell must be stronger to reach the energetic level of the hydrogen-rich–Si≡H_3_ groups via the valence band mechanism compared to the hydrogen-poor ≡Si–H groups. The averaged Si:H ratio of the affected Si–H_x_ groups shifted between examples 2 and 3 from 1:1.77 mol:mol (Si–H_1.77_) to 1:1.93 mol:mol (Si–H_1.93_), according to the overall balances shown in Figure 9b. The difference in the redox strength of the AuCl_4_^−^/Au half-cells is about 0.03 V (example 2: *E* (AuCl_4_^−^/Au) = 0.94 V vs. SHE vs. example 3: *E* (AuCl_4_^−^/Au) = 0.97 V vs. SHE). Including the four analytical findings between examples 2 and 3 in Figure 8e,f, the shift in the average H content in the attacked Si–H_x_ groups is about +0.05 mol per increase in the redox strength of the AuCl_4_^−^/Au half-cell by the amount of ≈+0.01 V vs. SHE.

### 3.4. Platinum Deposition onto Multicrystalline Silicon

Nine experiments were performed in the context of platinum deposition using the PtCl_6_^2−^ species as an oxidant. The initial molality of PtCl_6_^2−^ varied between *b* Pt (*diss., t* = 0 s) = 4.5 × 10^−7^ and 1.0 × 10^−2^ mol∙kg^−1^ at a HF level of 1.486 ± 0.016 mol∙kg^−1^. The redox strength of the PtCl_6_^2−^/Pt half-cell [95] ranged between *E* (PtCl_6_^2−^/Pt) = 0.65 and 0.72 V vs. SHE. The intermediate PtCl_6_^2−^/PtCl_4_^2−^ and PtCl_4_^2−^/Pt half-cells [28] were slightly weaker, with about *E* (PtCl_6_^2−^/PtCl_4_^2−^) = 0.48 to 0.60 V vs. SHE and *E* (PtCl_4_^2−^/Pt) = 0.55 to 0.68 V vs. SHE, respectively. Analogous to AuCl_4_^−^, the PtCl_6_^2−^ complex is also subject to pH-dependent ligand exchange with water [96], which will be neglected for the following considerations of the reaction processes.

Figure 10a shows the simplified energy diagram of the process of PtCl_6_^2−^ reduction on silicon. The PtCl_6_^2−^-induced valence transfer in the period of initial platinum deposition occurs via the valence band of the bulk silicon if the initial PtCl_6_^2−^ molality of *b* Pt *(diss., t* = 0 s) is < 6.0 × 10^−5^ mol∙kg^−1^ (*E* (PtCl_6_^2−^/Pt) < 0.68 V vs. SHE). In the case of an initial PtCl_6_^2−^ molality of ≥ 6.0 × 10^−5^ mol∙kg^−1^ (*E* (PtCl_6_^2−^/Pt) ≥ 0.68 V vs. SHE), the hydrogen-terminated silicon can also be attacked.

After the initial platinum deposition, the conduction and valence bands of the silicon are bent. The extent of band bending is the strongest due to the highest work function of platinum compared to the other materials studied [83,84,85,86,87,88]. The amount of valence band bending for the experiment with the lowest initial PtCl_6_^2−^molality of *b* Pt (*diss., t* = 0 s) = 4.5 × 10^−7^ mol∙kg^−1^ must have been at least +0.38 eV (*E_V_* (Si_bulk_) = −0.41 eV + 0.38 eV = −0.03 eV). Otherwise, the 2H_3_O^+^/H_2_ half-cell, which is weak at the platinum/silicon contact compared to the other metal/silicon configurations, with an equilibrium potential of *Eq. Pot.* (2H_3_O^+^/H_2_) = +0.03 V vs. SHE (*J* (Pt^4+^) = 1.1 × 10^−5^ mA∙cm^−2^) [26,33], would not have been able to reach the bulk silicon valence band.

According to the findings of our previous study [26] and this study, the oxonium-ion reduction is participating in the silicon oxidation process. Analogous to the behavior of AuCl_4_^−^ on gold, there is also probably the effect of an increasing negative dipole moment on the platinum surface with an increasing PtCl_6_^2−^ molality, from which follows an increase in the effective work function of the platinum. This assumption is based on our previous study’s findings [26] and that of Gossenberger et al. [92]. According to our previous study [26], there is an inflection point in the kinetics of platinum deposition at *b* Pt (*diss., t* = 0 s) = 5.1 × 10^−5^ mol∙kg^−1^, after which, first abruptly and then successively, the platinum deposition slows down at higher initial PtCl_6_^2−^ molalities. At the threshold, the energetic level of the valence band of the bulk silicon at the platinum/silicon contact matches the level of the Fermi energy, and the kinetics of platinum deposition is maximum [26]. Above this value, a Schottky barrier forms and the valence transfer to the valence band of the hydrogen-terminated silicon is favored. Since the kinetics of platinum deposition successively slows down further with increasing PtCl_6_^2−^ molality, the Schottky barrier behavior also seems to partially occur for hydrogen-terminated silicon (with lower hydrogen content). Kuznetsov et al. have evidenced a Schottky barrier height of +0.30 eV in an HF solution at the platinum/silicon contact [50].

It was further found in our previous study that oxonium-ion reduction is stopped at *b* Pt (*diss., t* = 0 s) = 6.0 × 10^−5^ mol∙kg^−1^ [26]. At this point, the 2H_3_O^+^/H_2_ half-cell has an equilibrium potential of *Eq. Pot.* (2H_3_O^+^/H_2_) = 0.00 V vs. SHE (*J* (Pt^4+^) = 2.1 × 10^−3^ mA∙cm^−2^) [26,33]. The valence transfer initiated by the 2H_3_O^+^/H_2_ half-cell must still occur via the valence band of the bulk silicon, i.e., via the tunneling effect through the Schottky barrier. Assuming an energetic level of the valence band of bulk silicon of *E_V_* (Si_bulk_) = +0.03 eV, the energetic level of the hydrogen-terminated silicon would be *E_V_* (Si-H_x_) *=* −0.24 eV (= −0.68 eV + 0.44 eV) for the same amount of band bending (0.03–−0.41 eV = +0.44 eV). Consequently, a reaction of the 2H_3_O^+^/H_2_ half-cell with the hydrogen-terminated silicon can be excluded. Above the threshold of *b* Pt (*diss., t* = 0 s) = 6.0 × 10^−5^ mol∙kg^−1^, a small overvoltage at a maximum of *Eq. Pot*. (2H_3_O^+^/H_2_) = −0.02 V vs. SHE (*J* (Pt^4+^) = 3.2 × 10^−1^ mA∙cm^−2^) occurs at *b* Pt (*diss., t* = 0 s) = 1.0 × 10^−2^ mol∙kg^−1^ [26,33]. This small overvoltage is sufficient to inhibit the oxonium-ion reduction [26].

Based on these preliminary considerations, the results of the platinum deposition shown in Figure 10b and the silicon dissolution plotted in Figure 10c have been divided into the participation of oxonium ions (red) and without the participation of oxonium ions in the oxidation process (black). It is noticeable that the platinum deposition (Δ*n* Pt *(diss., t* = 14,400 s)) was incomplete during the participation of the oxonium-ion reduction, even after the period of *t* = 14,400 s. The amount of silicon dissolution (Δ*n* Si (*diss., t* = 14,400 s)), in contrast, is about the same level as for silver and gold deposition, between Δ*n* Si (*diss., t* = 14,400 s) = 1.6 × 10^−6^ mol∙kg^−1^ at *b* Pt (*diss., t* = 0 s) = 4.5 × 10^−7^ mol∙kg^−1^ and Δ*n* Si (*diss., t* = 14,400 s) = 6.9 × 10^−6^ mol∙kg^−1^ at *b* Pt (*diss., t* = 0 s) = 5.2 × 10^−6^ mol∙kg^−1^.

This indicates that the oxonium ion-reduction at the platinum/silicon contact competes very strongly with the process of PtCl_6_^2−^ reduction. This is plausible because the energetic level of the 2H_3_O^+^/H_2_ half-cell is close to the level of the Fermi energy, and, thus, the valence transfer to the valence band of silicon is fast. Above the threshold of the oxonium-ion reduction at *b* Pt (*diss., t* = 0 s) = 6.0 × 10^−5^ mol∙kg^−1^, the PtCl_6_^2−^ reduction occurred without competing for oxonium-ion reduction, resulting in complete conversion to Pt within the period of *t* = 14,400 s. In the case of the silicon dissolution, the difference between the process with and without the contribution of the oxonium ions can be identified by the different slope of Δ*n* Si (*diss., t* = 14,400 s) in relation to *b* Pt (*diss., t* = 0 s) (Figure 10c).

In contrast to the other metal depositions, molecular hydrogen was not detectable in the gas phase in any of the nine experiments for platinum deposition (Δ*n* H_2_ (*g, t* = 14,400 s) < 1.8 × 10^−6^ mol). Gorostitza et al. [27] also noted the absence of molecular hydrogen formation at the platinum/silicon contact.

The comparison of the stoichiometric ratios from platinum and silicon dissolution (Δ*n* Pt*:*Δ*n* Si (*diss., t* = 14,400 s)) in Figure 10d shows, similar to the other metal depositions, a similar sigmoidal progression of the Δ*n* Pt*:*Δ*n* Si (*diss., t* = 14,400 s) ratios (minima and maxima functions according to [26]) in relation to *b* Pt *(diss., t* = 0 s). The stoichiometric Δ*n* Pt:Δ*n* Si (*diss., t* = 14,400 s) ratios at *b* Pt (*diss., t* = 0 s) ≤ 6.8 × 10^−7^ mol∙kg^−1^ are at 0.01:1 mol:mol, increase beyond that and reach a level of ≈1.05:1 mol:mol at *b* Pt (*diss., t* = 0 s) ≥ 1.0 × 10^−3^ mol∙kg^−1^.

The apparent paradox between the findings of the stoichiometric Δ*n* Pt*:*Δ*n* Si *(diss., t* = 14,400 s) ratio, indicating the partial participation of oxonium ions, and the lack of molecular hydrogen formation can be explained by the findings for examples 1 (highlighted in red) and 2 (highlighted in blue) indicated in Figure 10d.

The initial platinum deposition occurs at a redox strength of the PtCl_6_^2−^/Pt half-cell of *E* (PtCl_6_^2−^/Pt) = 0.65 V vs. SHE at *b* Pt (*diss., t* = 0 s) = 4.5 × 10^−7^ mol∙kg^−1^ (example 1) in interaction with the bulk silicon. The valence band of the hydrogen-terminated silicon is not initially accessible (*E_V_* (Si-H_x_) = 0.68 eV [82], Figure 10a). The reduction of PtCl_6_^2−^ occurs via the intermediate PtCl_4_^2−^ species to Pt (metal state). In this process, two holes are transferred twice to the valence band of the bulk silicon (Pt(IV) → Pt(II) + 2 h^+^ and Pt(II) → Pt + 2 h^+^). As a result, two Si–Si bonds are broken without forming dangling bonds (Figure 11a). This leads to the formation of either four Si^+^ intermediates or two Si^+^ intermediates and one Si^2+^ intermediate. Fluoride attaches to the site of the former Si–Si bonds. The resulting Si–F bonds destabilize the neighboring Si^+^–Si bonds, which are subsequently broken by H_2_O. OH^−^ binds on the side of the oxidized Si intermediates, and a Si–H bond is formed on the other side. The oxidized Si^z+^ species are dissolved by F^−^ or other anionic HF species to produce SiF_6_^2−^ in consecutive reactions (Figure 11a). In this respect, the course of the reaction corresponds to the formation of copper nuclei in Figure 7a. The difference between the platinum and copper deposition is that the valence transfer penetrates spatially deeper into the bulk silicon due to the stronger PtCl_6_^2−^/PtCl_4_^2−^/Pt half-cells compared to the Cu^2+^/Cu half-cell. An intermediate HSiF_3_ formation by the oxidation of the Si–Si back bonds of the ≡Si–H, =Si=H_2_ and–Si≡H_3_ groups, as outlined in Figure 7b, apparently does not occur in the case of platinum deposition. As a result, no molecular hydrogen will be generated.

The PtCl_6_^2−^/Pt half-cell with *E* (PtCl_6_^2−^/Pt) = 0.70 V vs. SHE is sufficiently strong in example 2 with *b* Pt (*diss., t* = 0 s) = 1.0 × 10^−3^ mol∙kg^−1^ that the initial valence transfer can occur via the valence band of the hydrogen-terminated silicon. After the initial platinum deposition, the band bending at the platinum/silicon contact allows all PtCl_6_^2−^/PtCl_4_^2−^/Pt half-cells to oxidize the hydrogen-terminated silicon. The PtCl_6_^2−^ and PtCl_4_^2−^ reduction to the metal state causes tetravalent and divalent hole transfers. These kinds of valence transfers to the Si–H bonds always result in the formation of one Si^+^ intermediate and H^+^. In consecutive reactions with H_2_O and F^−^ (or other anionic HF species), SiF_6_^2−^ and H_3_O^+^ are produced (Figure 11b). The formation of molecular hydrogen is excluded.

After the initial platinum deposition, the oxonium-ion reduction starts if the initial PtCl_6_^2−^ molality is below the threshold of *b* Pt (*diss., t* = 0 s) = 6.0 × 10^−5^ mol∙kg^−1^, similar to example 1, but not example 2. According to the reaction scheme in Equation (2), this process would produce at least one hydrogen radical, leading to the formation of molecular hydrogen. Gorostitza et al. [27] postulated that the platinum nuclei would possibly induce the re-oxidation of the molecular hydrogen to hydrogen ions. Therefore, the emission of molecular hydrogen would be prevented. By contrast, however, platinum is known to be one of the best catalysts for H_2_ formation [97].

In view of this contradiction, an alternative theory is derived: The metals influence the number of valence transfers, starting from the oxonium-ion reduction via metal/silicon contact to the silicon. The metals that can be oxidized up to the oxidation state of +I (silver (Ag^+^), copper (Cu^+^) and gold (Au^+^)) cause a monovalent valence transfer to the silicon. This leads to the formation of hydrogen radicals, according to Equation (2). The next higher oxidation state above the metal state for platinum is +II [28,95]. This provokes a divalent valence transfer from the oxonium ion via the platinum/silicon contact to the silicon. Following Equation (13), this leads to the formation of an intermediate hydride (H^−^) on the cathodic side:(13)$$H_{3}O^{+}F^{-}\to H^{-}+H_{2}O\;\left(⇌\;\cdot H\;+\;\cdot \mathrm{OH}\right)+F^{-}+2{\;h}^{+}$$

On the anodic side, a Si–Si bond break occurs with the formation of two Si^+^ intermediates. One of the Si^+^ intermediates reacts consecutively with the hydride species, resulting in a Si–H bond. The other Si^+^ intermediate is converted by further oxidation steps by either H_3_O^+^ or H_3_O^+^ F^−^ and/or by H_2_O and F^−^ to the formal end product SiF_6_^2−^ without the formation of molecular hydrogen (Figure 11c).

The balances of the overall reactions of examples 1 and 2 are shown in Figure 11d. In example 1, the contribution of the oxonium ions to the silicon oxidation process is taken into account, which is significantly larger than the amount of the PtCl_6_^2−^ reduction to the metal state. In example 2, the oxonium-ion reduction is inhibited by the overvoltage effect of the 2H_3_O^+^/H_2_ half-cell on platinum. Considering the strength of the PtCl_6_^2−^/PtCl_4_^2−^/Pt half-cells, the reaction occurs with the ≡Si–H, =Si=H_2_ and–Si≡H_3_ groups and not with the bulk silicon. Based on the stoichiometric Δ*n* Pt*:*Δ*n* Si (*diss., t* = 14,400 s) ratio of ≈1.05:1 mol:mol, a slightly higher proportion of oxidation of more hydrogen-rich groups can be inferred by analogy with the conclusions regarding gold deposition. Otherwise, if the ≡Si–H, =Si=H_2_ and–Si≡H_3_ groups were oxidized in an equally distributed manner, a Δ*n* Pt:Δ*n* Si ratio of 1:1 mol:mol should have been observed.

## 4. Conclusions

The experiments in this study were carried out to elucidate the reaction processes behind the process of precious metal-ion reduction on silicon in a dilute HF matrix. The basic findings had already been elaborated on in our previous study [26]. Based on these, this research focused on the controversially discussed question regarding the formation of molecular hydrogen.

A total of 89 batch experiments were performed for its examination consisting of aqueous solutions with different additions of Ag^+^, Cu^2+^, AuCl_4_^−^ or PtCl_6_^2−^ and HF. Multicrystalline silicon wafers were transferred into these solutions. The HF solutions were sampled before the addition, and after *t* = 3600 s (Ag, Cu, Au) or *t* = 14,400 s (Pt) processing, and the metal deposition (Δ*n* Me, Me = Ag, Cu, Au or Pt) and silicon dissolution (Δ*n* Si) were quantified by ICP–OES analyses. The amount of molecular hydrogen formation (Δ*n* H_2_) was determined by mass spectrometric analysis of the supernatant gas phase. Possible reaction processes were discussed concerning the stoichiometric balances of Δ*n* Me, Δ*n* Si and Δ*n* H_2_, the information from kinetic and stoichiometric balances from the previous study [26], and the energetic consideration of the half-cells involved and the metal/silicon contacts, using selected examples and quantified in terms of overall reaction equations. As a result of this approach, the process of metal-ion reduction to silicon and molecular hydrogen formation can be characterized as follows:

The reduction of Ag^+^, AuCl_4_^−^ or PtCl_6_^2−^ starts on the entire hydrogen-terminated surface of the silicon. Depending on whether the redox potential of the Ag^+^/Ag, AuCl_4_^−^/Au or PtCl_6_^2−^/Pt half-cells is less or greater than 0.68 V vs. SHE, the valence transfer of one or more holes occurs via the valence band of either the bulk silicon (if *E* (Me^z+^/Me) < 0.68 V vs. SHE) or the hydrogen-terminated silicon (if *E* (Me^z+^/Me) ≥ 0.68 V vs. SHE). In the period of the initial nucleation, the Cu^2+^/Cu and the Cu^2+^/Cu^+^ half-cells are too weak to reach the energetic level of the valence band of the bulk silicon or that of the hydrogen-terminated silicon. The initial reduction of Cu^2+^ to Cu requires the existence of disordered sites where non-hydrogen-terminated silicon is present on the surface, which can be attacked by the Cu^2+^/Cu^+^/Cu half-cells. As soon as metal nucleation has occurred, the band bending of the conduction and valence bands of the silicon at the metal/silicon contact results. The amount of band bending depends on the difference between the work function of the metal and the hydrogen-terminated silicon [26]. In the case of gold and platinum deposition, the amount of band bending is affected by the concentration of the anionic metal complexes AuCl_4_^−^ and PtCl_6_^2−^, respectively. These species can change the metals’ effective work functions by their negative dipole moment at the interface between the metal surface and the HF solution.

The amount of band bending and the strength of the Ag^+^/Ag, Cu^2+^/Cu, AuCl_4_^−^/Au or PtCl_6_^2−^/Pt half-cells influence whether further valence transfer occurs via the valence band of the bulk silicon or the hydrogen-terminated silicon. The metal ion/metal system determines the number of valence transfers. Both factors determine whether and how molecular hydrogen is formed or not.

The reduction of Ag^+^ to Ag results in a monovalent hole transfer (1 h^+^), and that of AuCl_4_^−^ to Au results in a trivalent charge transfer (3 h^+^). When an odd number of holes are transferred to the valence band of the bulk silicon, Si–Si bonds are broken, forming Si^+^ intermediates and dangling bonds (●Si). Water (or possibly also HF) is suspected to react with the dangling bonds, creating hydrogen radicals (∙H). Hydrogen radicals combine with each other to form molecular hydrogen. The intermediate Si^+^ will be further oxidized by H_2_O and several HF species. If instead of this process, Ag^+^/Ag or AuCl_4_^−^/Au half-cells transfer an odd-number of valences to the bulk silicon again, redundant dangling bonds will be eliminated and, consequently, no molecular hydrogen is formed by consecutive reactions.

If the reaction of the Ag^+^/Ag or AuCl_4_^−^/Au half-cells occurs with the hydrogen-terminated silicon, hydrogen formation is the consequence of a monovalent bond break of the Si–H bonds with intermediate hydrogen radical formation. By contrast, it does not happen if two valences are cumulatively transferred to the Si–H bonds. In this case, a hydrogen ion is formed and, in a subsequent reaction with water, an oxonium ion. In principle, both the odd-numbered and the cumulative even-numbered valence transfers occur during silver and gold deposition. The lower the metal ion concentration, the more the odd-numbered valence transfers and, thus, the formation of molecular hydrogen predominates in relation to the silicon oxidation and its consecutive dissolution. Conversely, the even-numbered valence transfers dominate, and the molecular hydrogen formation decreases in higher concentrated metal ion solutions. According to the analyses, the ratio of Δ*n* H_2_*:*Δ*n* Si varies between ≈1.2:1 and ≈0.01:1 mol:mol for silver deposition and between ≈0.8:1 and ≈0.18:1 mol:mol for gold deposition.

With the reduction of PtCl_6_^2−^ via the intermediate PtCl_4_^2−^ to Pt, there can only be divalent or, when skipping the intermediate PtCl_4_^2−^ species, tetravalent valence transfers to the bulk silicon or hydrogen-terminated silicon. Consequently, neither dangling bonds are formed during Si–Si bond-breaking nor hydrogen radicals during the oxidation of Si–H bonds. Therefore, platinum deposition always proceeds without the formation of molecular hydrogen.

The Cu^2+^/Cu half-cell at the copper/silicon contact is sufficiently strong to initiate divalent (2 h^+^) and tetravalent (2 × 2 h^+^) valence transfers via the valence band of the bulk silicon. Whereas in silver, gold and platinum deposition (cumulative), even-numbered valence transfers cannot lead to a formation of molecular hydrogen, this is possible in the copper deposition process. The Cu^2+^/Cu half-cell is weaker than the Ag^+^/Ag, AuCl_4_^−^/Au, PtCl_6_^2−^/Pt and its intermediate half-cells. Therefore, the valence transfer occurs spatially shallower into the bulk silicon, leading to a partial attack on the Si–Si back bonds of the surface ≡Si–H, =Si=H_2_ and–Si≡H_3_ groups. As a result, the intermediate HSiF_3_ is formed, which is subsequently converted to H_2_ and SiF_6_^2−^ in solution, resulting in a Δ*n* H_2_*:*Δ*n* Si ratio of 1:1 mol:mol. The stoichiometric Δ*n H_2_:*Δ*n Si* ratios determined are between 0.42:1 and 0.16:1 mol:mol without enabled oxonium-ion reduction (*b* Cu (*diss., t* = 0 s) > 2.5 × 10^−4^ mol∙kg^−1^) and without the direct oxidation of the surface Si–H bonds (*b* Cu (*diss., t* = 0 s) < 8.2 × 10^−3^ mol∙kg^−1^) [26]. This result implies that the non-H_2_-forming reactions are in the majority and increase with increasing initial Cu^2+^ molality.

Monovalent valence transfers through the formation of intermediate Cu^+^ are also possible in copper deposition, but only under specific conditions. The Cu^2+^/Cu^+^ half-cell is too weak to transfer holes via the valence band of the bulk and hydrogen-terminated silicon. Any reaction by this half-cell is only possible at interfering sites with surface-exposed, non-hydrogen-terminated silicon. However, Cu^+^ can be formed by a comproportionation reaction of Cu^2+^ with the metallic copper layer already deposited on the silicon. The process is not favored energetically but can become relevant if, during Cu^2+^ reduction to Cu, the valence transfer to the bulk silicon is inhibited by forming a layer of oxidized silicon acting as an insulator. This is feasible if the anionic HF dissociation species F^−^, HF_2_^−^ and H_2_F_3_^−^, responsible for rapid silicon dissolution, have a sufficiently low activity [26]. The Cu^+^/Cu half-cell with a redox level of *E* (Cu^+^/Cu) ≥ 0.38 V vs. SHE is strong enough to transfer monovalent valences to the Si–H bonds of the surface ≡Si–H, =Si=H_2_ and–Si≡H_3_ groups. This results in the formation of hydrogen radicals and subsequently in the formation of molecular hydrogen. The higher the concentration of Cu^+^, the more the percentage of cumulative even-numbered valence transfers from the Cu^+^/Cu half-cell to the ≡Si–H, =Si=H_2_ and–Si≡H_3_ groups increases. Consequently, less molecular hydrogen is formed with increasing Cu^+^ molality. The Δ*n* H_2_*:*Δ*n* Si ratio at *b* Cu (*diss., t* = 0 s) = 8.2 × 10^−3^ mol∙kg^−1^ was about 0.16:1 mol:mol and decreased to half by *b* Cu (*diss., t* = 0 s) = 1.5 × 10^−1^ mol∙kg^−1^.

If metal deposition has initially occurred on the silicon, oxonium-ion reduction (or H_3_O^+^ F^−^ ion-pair reduction) can start as a second cathodic process at the metal/silicon contact. The basic condition for this process is that an underpotential effect of the 2H_3_O^+^/H_2_ half-cell appears on the metal surface. This effect must be so significant that the resulting equilibrium potential of the 2H_3_O^+^/H_2_ half-cell reaches at least the energetic level of the valence band of the bulk silicon. The level of the equilibrium potential of the 2H_3_O^+^/H_2_ half-cell depends on the current density of the metal ions on the metal/silicon surface [33] and, thus, corresponds to the molality of the metal ions [26]. For oxonium-ion reduction to proceed, *b* Ag (*diss., t* = 0 s) must not exceed 3.3 × 10^−4^ mol∙kg^−1^. The threshold values for the other metal depositions are *b* Cu (*diss., t* = 0 s) = 2.5 × 10^−4^ mol∙kg^−1^, *b* Au *(diss., t* = 0 s) = 1.5 × 10^−4^ mol∙kg^−1^, and *b* Pt (*diss., t* = 0 s) = 6.0 × 10^−5^ mol∙kg^−1^ [26].

Oxonium-ion reduction leads to the formation of molecular hydrogen at the silver/silicon, copper/silicon and gold/silicon contacts. No molecular hydrogen is produced at the platinum/silicon contact.

The metals, which can undergo the oxidation state of +I upon their oxidation, catalyze a monovalent valence transfer from the oxonium ion to silicon, resulting in the formation of hydrogen radical and water and a ≡Si–F intermediate generation. Water resulting from the oxonium-ion reduction subsequently reacts with a neighboring Si–Si bond destabilized by the Si–F bond. The further reaction path depends on the metal/silicon contact. As a general rule, the lower the highest charge number of the respective metal ion, the richer the hydrogen formation of this process. Consequently, in the case of silver (Ag^+^), more hydrogen radicals are formed in the reaction with the bulk silicon and more molecular hydrogen in the subsequent reactions compared to copper (Cu^2+^) and gold (Au^3+^). On average, the oxonium-ion reduction process with the bulk silicon leads to a Δ*n* H_2_*:*Δ*n* Si ratio of ≈3:4 mol:mol for silver deposition and ≈1:2 mol:mol for copper deposition.

In the case of the gold deposition, there is also the reaction with the Si–H bonds in addition to the reaction with the Si–Si bonds of the bulk silicon. During this process, one hydrogen radical is formed due to the reduction of the oxonium ion and a second is formed due to the bond breakage of Si–H. This results in a Δ*n* H_2_:Δ*n* Si ratio of 1:1 mol:mol. The mixture of the reaction of the oxonium ion with the bulk silicon and the hydrogen-terminated silicon results in a Δ*n* H_2_:Δ*n* Si ratio of ≈0.6:1 mol:mol.

The oxonium-ion reduction in platinum deposition follows a different reaction course. The next higher oxidation state of platinum is +II. Related to this fact, the transfer of two holes to a Si–Si bond in bulk silicon occurs at the platinum/silicon contact, resulting in the formation of two Si^+^ intermediates. The oxonium ion is converted to an intermediate hydride and water. The hydride reacts with one of the Si^+^ intermediates to form a Si–H bond. The other Si^+^ intermediate reacts further with oxonium ions and/or water and fluoride up to the formal end product SiF_6_^2−^. The Si–H bonds themselves are not accessible by the oxonium ions at the platinum/silicon contact. The oxonium-ion reduction in platinum deposition with the bulk silicon is kinetically favored compared to the other metal/silicon systems because the equilibrium potential of the 2H_3_O^+^/H_2_ half-cell is approximately equal to the Fermi energy at the platinum/silicon contact and, simultaneously, also the valence band of the bulk silicon. This leads to the fact that oxonium-ion reduction slows down further platinum deposition after the initial platinum deposition [26]. Since no monovalent valence transfers occur in either process, the Δ*n* H_2_*:*Δ*n* Si ratio is an unexceptional 0:1 mol:mol.

Finally, Table 1 gives a comparative overview of the discussed reaction processes, the range of the initial metal ion molalities in which the respective reaction processes occur, and the respective analytically derived Δ*n* H_2_*:*Δ*n* Si ratios.

## Figures and Tables

**Figure 1 nanomaterials-11-00982-f001:**
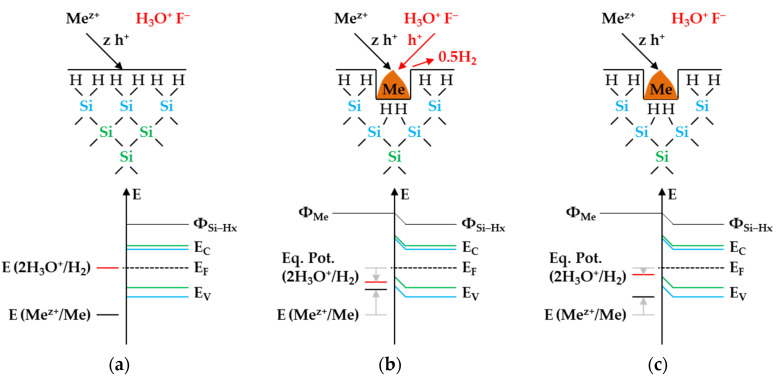
Simplified reaction and energy schemes of the metal ion/metal (Me^z+^/Me) and 2H_3_O^+^/H_2_ half-cells in the presence of silicon, showing the energetic levels of the valence and conduction bands of the bulk silicon (green) and the hydrogen-terminated silicon (blue) (**a**) before metal deposition, (**b**) after initial metal deposition at the metal/silicon contact with an equilibrium potential of the 2H_3_O^+^/H_2_ half-cell (*Eq. Pot.* (2H3O^+^/H_2_)) in the range of the energy level of the silicon valence bands and (**c**) out of the energy level of the silicon valence bands.

**Figure 2 nanomaterials-11-00982-f002:**
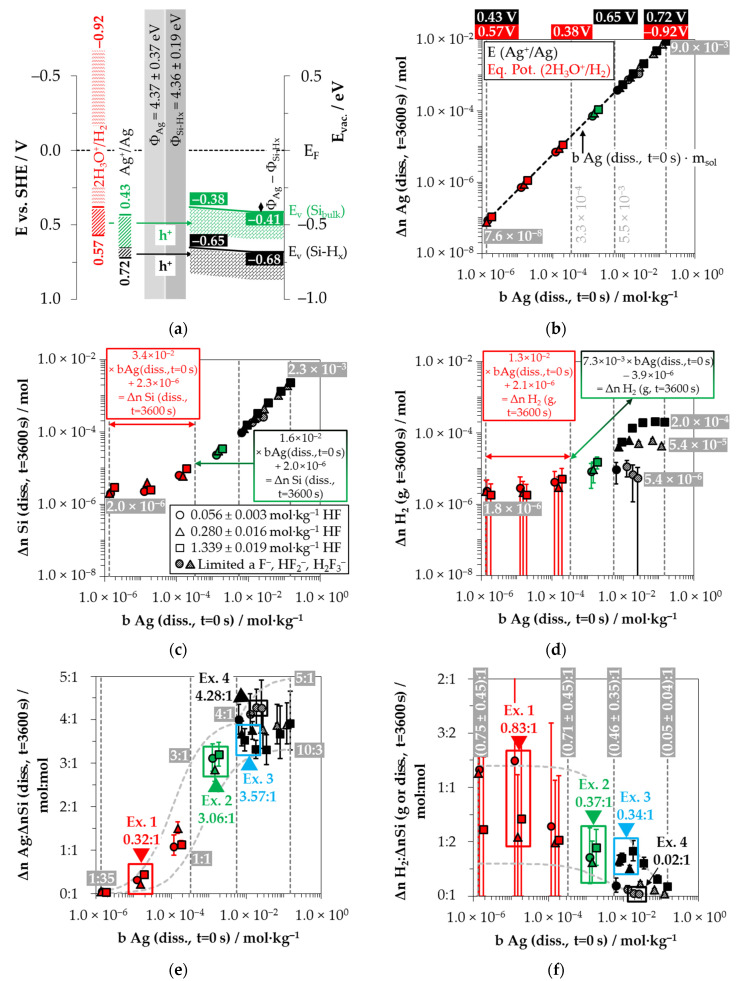
(**a**) Energy diagram of Ag^+^ and H_3_O^+^ reduction at the Ag/Si contact with the indication of the upper and lower limits of the redox strengths of the Ag^+^/Ag half-cell [28] and those of the equilibrium potentials of the 2H_3_O^+^/H_2_ half-cell [33] and energetic levels of the valence bands of the bulk and hydrogen-terminated silicon (*E_V_* (Si_bulk_) [26,80,81], *E_V_* (Si-H_x_) [82]) and their bending, based on the differences of *Φ*_Ag_ and *Φ*_Si-Hx_ [83,84,85,86,87,88] and the findings from [26]; (**b**) amounts of Ag deposition (Δ*n* Ag (*diss., t* = 3600 s)); (**c**) Si dissolution (Δ*n* Si (*diss., t* = 3600 s)); and (**d**) molecular H_2_ formation (Δ*n* H_2_ (*g, t* = 3600 s)) after *t* = 3600 s processing, as well as stoichiometric ratios of (**e**) Ag deposition and Si dissolution (Δ*n* Ag*:*Δ*n* Si *(diss., t* = 3600 s)); and (**f**) stoichiometric ratios of molecular H_2_ formation and Si dissolution (Δ*n* H_2_*:*Δ*n* Si *(g* or *diss., t* = 3600 s)) relative to the initial Ag^+^ molality *b* Ag (*diss., t* = 0 s).

**Figure 3 nanomaterials-11-00982-f003:**
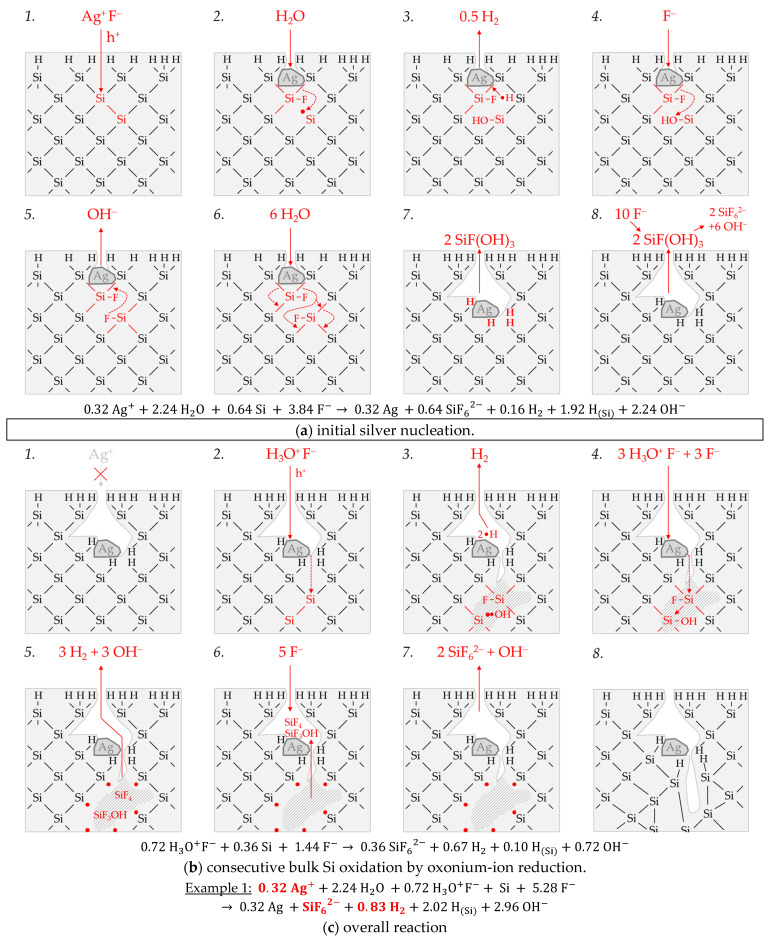
Schematic reaction process and balance, based on example 1 labeled in Figure 2e,f, with a Δ*n* Ag:Δ*n* Si*:*Δ*n* H_2_ ratio of ≈0.32:1:0.83 mol:mol:mol (**a**) with eight reaction steps of initial silver nucleation, (**b**) subsequent oxonium-ion reduction and (**c**) the balance of the overall reaction.

**Figure 4 nanomaterials-11-00982-f004:**
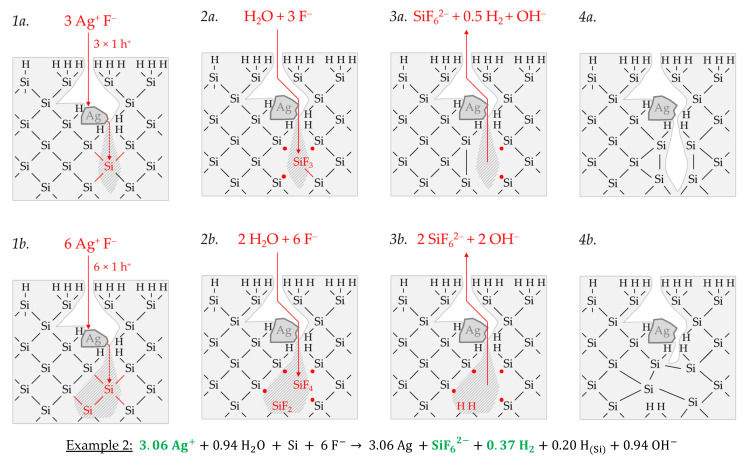
Schematic reaction process and balance of silver deposition on silicon, based on example 2 labeled in Figure 2e,f, with a Δ*n* Ag:Δ*n* Si:Δ*n* H_2_ ratio of ≈3.06:1:0.37 mol:mol:mol.

**Figure 5 nanomaterials-11-00982-f005:**
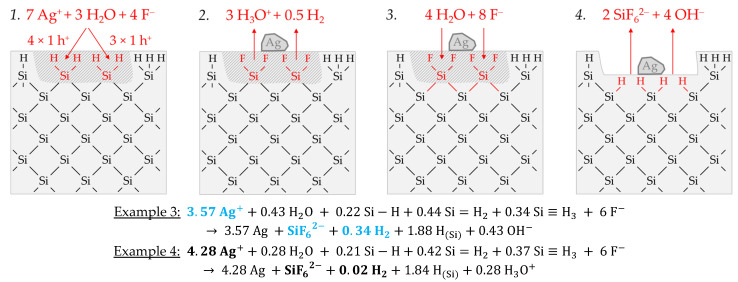
Schematic reaction process and balance of the silver deposition on silicon, based on examples 3 and 4 labeled in Figure 2e,f, with a Δ*n* Ag*:*Δ*n* Si:Δ*n* H_2_ ratio of ≈ 3.57:1:0.34 and 4.28:1:0.02 mol:mol:mol, respectively.

**Figure 6 nanomaterials-11-00982-f006:**
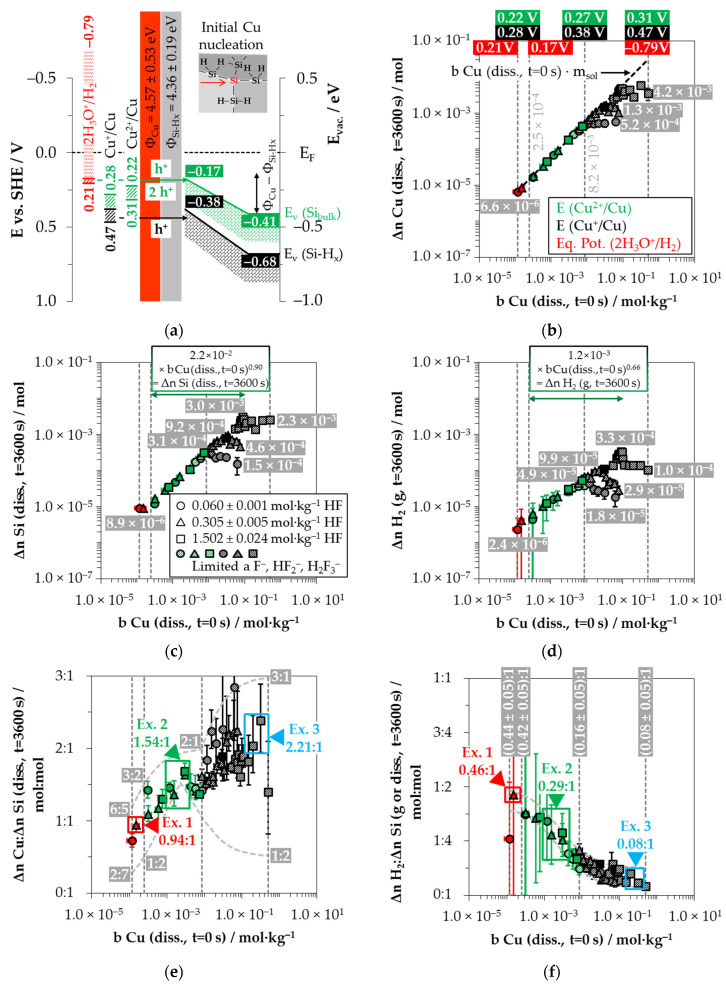
(**a**) Energy diagram of Cu^2+^/Cu^+^ and H_3_O^+^ reduction at the Cu/Si contact with the indication of the upper and lower limits of the redox strengths of the Cu^2+^/Cu and Cu^+^/Cu half-cells [28] and those of the equilibrium potentials of the 2H_3_O^+^/H_2_ half-cell [33] and energetic levels of the valence bands of the bulk and hydrogen-terminated silicon (*E_V_* (Si_bulk_) [26,80,81], *E_V_* (Si-H_x_) [82]) and their bending based on the differences of *Φ*_Cu_ and *Φ*_Si-Hx_ [83,84,85,86,87,88] and the findings from [26], (**b**) amounts of Cu deposition (Δ*n* Cu (*diss., t* = 3600 s)); (**c**) Si dissolution (Δ*n* Si (*diss., t* = 3600 s)) and (**d**) molecular H_2_ formation (Δ*n* H_2_ (*g, t* = 3600 s)) after *t* = 3600 s processing, as well as stoichiometric ratios of (**e**) Cu deposition and Si dissolution (Δ*n* Cu*:*Δ*n* Si (*diss., t* = 3600 s)) and (**f**) stoichiometric ratios of molecular H_2_ formation and Si dissolution (Δ*n* H_2_*:*Δ*n* Si (*g* or *diss., t* = 3600 s)) relative to the initial Cu^2+^ molality *b* Cu (*diss., t* = 0 s).

**Figure 7 nanomaterials-11-00982-f007:**
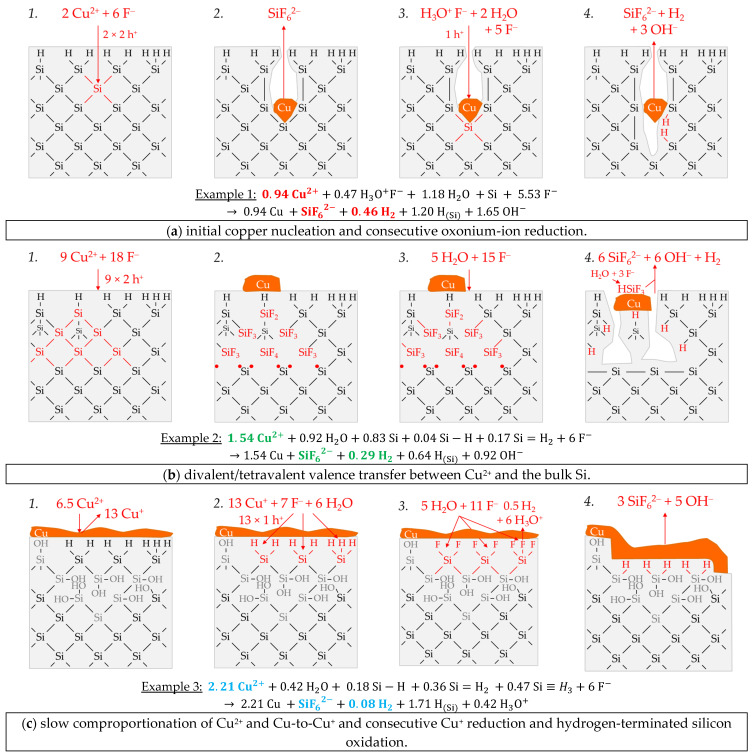
Schematic reaction process and balance of the copper deposition on silicon, based on the examples marked in Figure 6e,f. (**a**) Example 1: the copper nucleation and consecutive oxonium-ion reduction, (**b**) example 2: the valence exchange between Cu^2+^ and the bulk Si, and (**c**) example 3: the valence exchange between Cu^+^ and the hydrogen-terminated silicon.

**Figure 8 nanomaterials-11-00982-f008:**
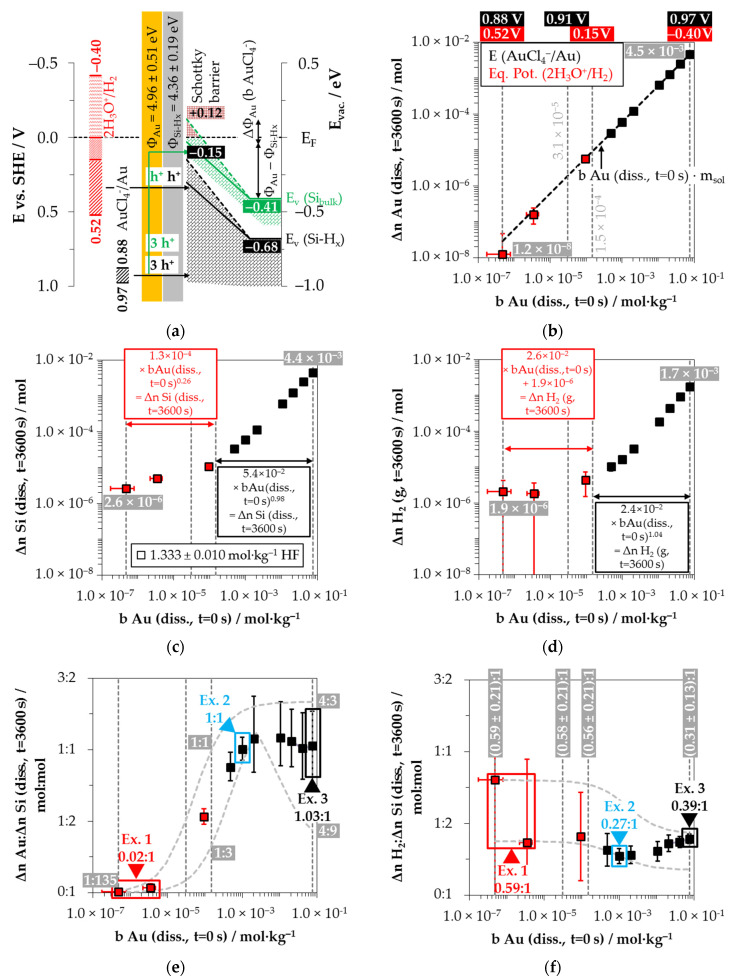
(**a**) Energy diagram of AuCl_4_^−^ and H_3_O^+^ reduction at the Au/Si contact with the indication of the upper and lower limits of the redox strengths of the AuCl_4_^−^/Au half-cells [28] and those of the equilibrium potentials of the 2H_3_O^+^/H_2_ half-cell [33] and energetic levels of the valence bands of the bulk and hydrogen-terminated silicon (*E_V_* (Si_bulk_) [26,80,81], *E_V_* (Si-H_x_) [82]) and their bending based on the differences of *Φ*_Au_ and *Φ*_Si-Hx_ [83,84,85,86,87,88] and the findings from [26]; (**b**) amounts of Au deposition (Δ*n* Au (*diss., t* = 3600 s)), (**c**) Si dissolution (Δ*n* Si (*diss., t* = 3600 s)) and (**d**) molecular H_2_ formation (Δ*n* H_2_ (*g, t* = 3600 s)) after *t* = 3600 s processing, as well as stoichiometric ratios of (**e**) Au deposition and Si dissolution (Δ*n* Au*:*Δ*n* Si *(diss., t* = 3600 s)) and (**f**) stoichiometric ratios of molecular H_2_ formation and Si dissolution (Δ*n* H_2_*:*Δ*n* Si (*g* or *diss., t* = 3600 s)) relative to the initial AuCl_4_^−^ molality *b* Au (*diss., t* = 0 s).

**Figure 9 nanomaterials-11-00982-f009:**
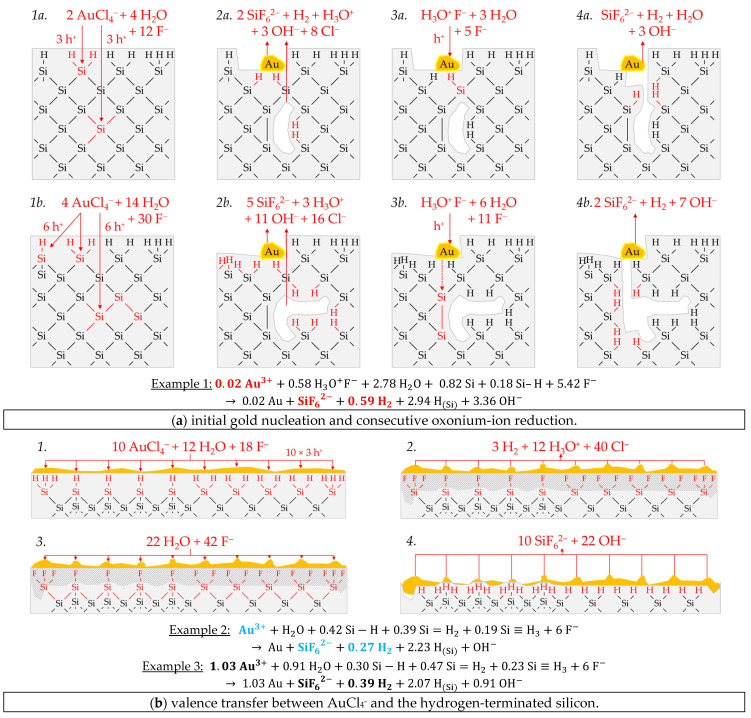
Schematic reaction process and balance of the gold deposition on silicon, based on the examples marked in Figure 8e,f. (**a**) Example 1: the gold nucleation and consecutive oxonium-ion reduction, (**b**) examples 2 and 3: the valence exchange between AuCl_4_^−^ and the hydrogen-terminated silicon.

**Figure 10 nanomaterials-11-00982-f010:**
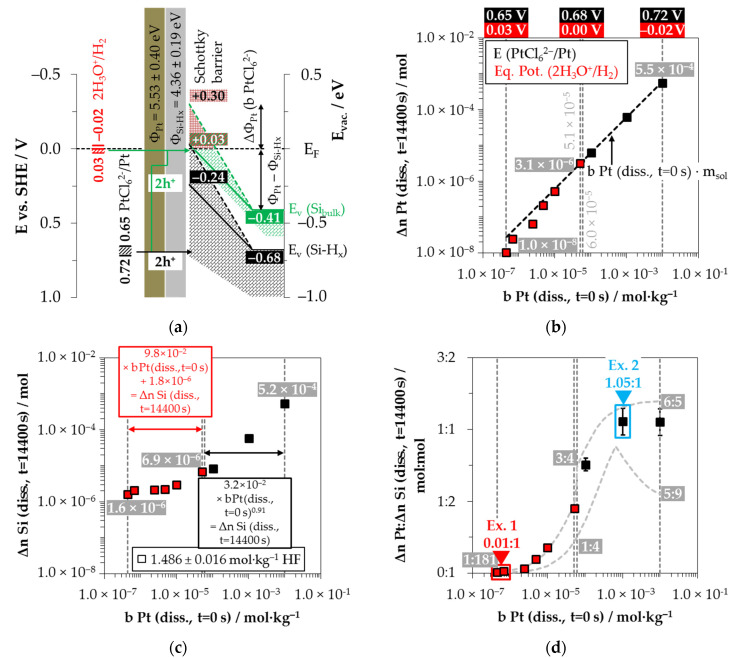
(**a**) Energy diagram of PtCl_6_^2−^ and H_3_O^+^ reduction at the Pt/Si contact with the indication of the upper and lower limits of the redox strengths of the PtCl_6_^2−^/Pt half-cell [95] and those of the equilibrium potentials of the 2H_3_O^+^/H_2_ half-cell [33], and energetic levels of the valence bands of the bulk and hydrogen-terminated silicon (*E_V_* (Si_bulk_) [26,80,81], *E_V_* (Si-H_x_) [82]) and their bending, based on the differences of *Φ*_Pt_ and *Φ*_Si-Hx_ [83,84,85,86,87,88] and the findings from [26]; (**b**) amounts of Pt deposition (Δ*n* Pt (*diss., t* = 14,400 s)) and (**c**) Si dissolution (Δ*n* Si (*diss., t* = 14,400 s)), as well as stoichiometric ratios of (**d**) Pt deposition and Si dissolution (Δ*n* Pt:Δ*n* Si (*diss., t* = 14, 400 s)) relative to the initial PtCl_6_^2−^ molality *b* Pt (*diss., t* = 0 s); Δ*n* H_2_ (*g, t* = 14,400 s) < 1.8 × 10^−6^ mol in each case.

**Figure 11 nanomaterials-11-00982-f011:**
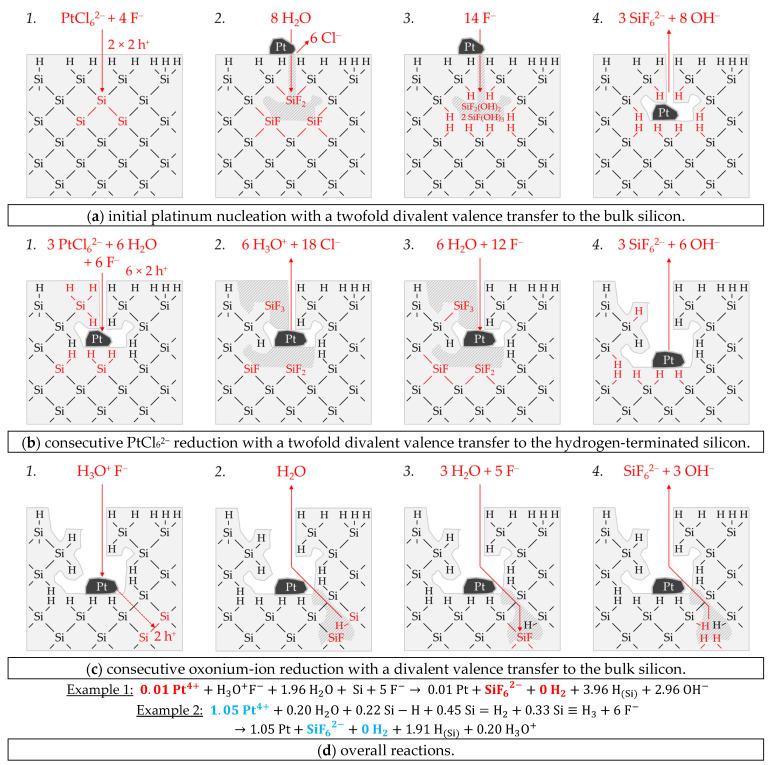
Schematic reaction process and balance of the platinum deposition on silicon, based on the examples marked in Figure 10e,f, showing (**a**) the initial platinum deposition, (**b**) the consecutive PtCl_6_^2−^ reduction, (**c**) the consecutive oxonium-ion reduction if *Eq. Pot.* (2H_3_O^+^/H_2_) ≥ 0 V vs. SHE and (**d**) balances of the overall reactions of examples 1 and 2.

**Table 1 nanomaterials-11-00982-t001:** Reaction processes and their range in relation to the initial metal ion molalities *b* Me *(diss., t* = 0 s) and the analytically derived average Δ*n* H_2_:Δ*n* Si ratios.

Reaction Process	Ag^+^/Ag	Cu^2+^/Cu	AuCl_4_^−^/Au	PtCl_6_^2−^/Pt
H_3_O^+^ F^−^ reduction after initial metal nucleation	*b* Ag (*diss., t* = 0 s) ≤ 3.3 × 10^−4^ mol∙kg^−1^	*b* Cu (*diss., t* = 0 s) ≤ 2.5 × 10^−4^ mol∙kg^−1^	*b* Au (*diss., t* = 0 s) ≤ 1.5 × 10^−4^ mol∙kg^−1^	*b* Pt (*diss., t* = 0 s) ≤ 0.6 × 10^−4^ mol∙kg^−1^
	Δ*n* H_2_:Δ*n* Si ≈ 0.75:1–0.71:1 mol:mol	Δ*n* H_2_:Δ*n* Si ≈ 0.44:1–0.42:1 mol:mol	Δ*n* H_2_:Δ*n* Si ≈ 0.59:1–0.56:1 mol:mol	Δ*n* H_2_:Δ*n* Si = 0:1 mol:mol
Me^z+^ reduction vs. bulk Si oxidation	*b* Ag (*diss., t* = 0 s) > 3.3 × 10^−4^– 5.5 × 10^−3^ mol∙kg^−1^	*b* Cu (*diss., t* = 0 s) > 2.5 × 10^−4^– 8.2 × 10^−3^ mol∙kg^−1^		
	Δ*n* H_2_:Δ*n* Si ≈ 0.71:1–0.46:1 mol:mol	Δ*n* H_2_:Δ*n* Si ≈ 0.42:1–0.16:1 mol:mol		
Me^z+^ reduction vs. bulk Si and Si–H_x_ oxidation	*b* Ag (*diss., t* = 0 s) > 5.5 × 10^−3^– 1.5 × 10^−1^ mol∙kg^−1^	*b* Cu (*diss., t* = 0 s) > 8.2 × 10^−3^– 1.5 × 10^−1^ mol∙kg^−1^		
	Δ*n* H_2_:Δ*n* Si ≈ 0.46:1–0.05:1 mol:mol	Δ*n* H_2_:Δ*n* Si ≈ 0.16:1–0.08:1 mol:mol		
Me^z+^ reduction vs. Si–H_x_ oxidation			*b* Au (*diss., t* = 0 s) > 1.5 × 10^−4^– 7.6 × 10^−3^ mol∙kg^−1^	*b* Pt (*diss., t* = 0 s) > 0.6 × 10^−4^– 1.0 × 10^−2^ mol∙kg^−1^
			Δ*n* H_2_:Δ*n* Si ≈ 0.56:1–0.27:1 mol:mol	Δ*n* H_2_:Δ*n* Si = 0:1 mol:mol

## Data Availability

The data presented in this study are available on request from the corresponding author.

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
