# Peer review of "The Role of the Molecular Hydrogen Formation in the Process of Metal-Ion Reduction on Multicrystalline Silicon in a Hydrofluoric Acid Matrix"

_nanomaterials, 2021, doi:10.3390/nano11040982_

Round 1

Reviewer 1 Report

Stefan et al. comprehensively investigated the role of Hin the process of metal ion reduction on silicon in HF.  I would say it’s an excellent study which gives a detailed information on the effect of H on the deposition of Ag, Cu, Au and Pt. The experiments are designed very well, and the obtained results are well supported by the results. The authors tried their best to make the paper attractive and understandable to the readers. Because their conclusions are well supported by their results, I am well convinced to accept the paper for publication after minor comments.

  1. There are some topological errors, I would recommend the authors to proofread the paper.
  2. Due to the lengthy manuscript, it's difficult to gain the conclusions and experimental design performed in the work. I suggest some tabular form could help well to understand the result in a glace. Authors can prepare a comparative tabular form for the deposition parameters, amount of H2 formed, redox potentials mass spectroscopic results, and other important distinguishable points in each case of depositon.

                                              ***The end***

Author Response

Dear reviewer,

Thank you for taking the time to critically review our study. We are pleased that we were able to convince you with our research.

According to your first remark, we have read our manuscript again thoroughly. We hope that we have eliminated all the minor errors.

We have responded to your second comment by creating a clear and simple table in the conclusion. The table shows the reaction processes and their range of values with reference to the metals. In accordance with the focus of the study, the stoichiometries of the molecular hydrogen formation in relation to the silicon dissolution are also implemented. The underlying analyses and other parameters are incorporated in the figures 2, 6, 8 and 10 in detail. Therefore, we did not want to overload the summary table with further information.

Yours sincerely

Stefan Schönekerl and Jörg Acker

Reviewer 2 Report

The manuscript is well presented and provides interesting data about the molecular hydrogen formation during electroless deposition of metals on silicon substrates. A large batch of experiments has been carried out in order to bring insights on the hydrogen formation. Analytical quantification of hydrogen during the deposition has been achieved, which is nice and new in this specific area. Overall, the study is well written and properly presented. Some questions still need to be answered and some minor changes should be carried out prior to publication.

  • The study focused on the hydrogen generation taking place during the deposition of 4 different metals: Ag, Cu, Au and Pt. The authors should explain why they chose to study these metals in particular, for which applications the systems “Si/different metals” can be used, and why these 4 metals are the most appropriate to study hydrogen generation.

  • The authors focused their work on the study of cathodic reduction of hydrogen ions in electroless deposition process. A major addition to the introduction seems appropriate to bring the reader to understand why they applied this technique. It is important that the authors explain why they chose to study the deposition of metals on silicon using this route, and how electroless deposition compares with other deposition techniques more typically used in the semiconductor industry, such as PVD (physical vapor deposition), CVD (chemical vapor deposition) or ALD (atomic layer deposition). In fact, considering the deposition of metallic ultra-thin films on silicon, ALD appears as the technique of choice, and some work reporting on the nucleation of various noble metals by ALD should be cited (Leick et al, Journal of Physics D: Applied Physics 49 (11), 115504, 2016 ; Weber et al, The Journal of Physical Chemistry C 118 (16), 8702-8711 (2014)).

  • This study brings fundamental insights on hydrogen generation. Hydrogen is currently being studied as a potential fuel for transportation, and a paragraph should be added in the introduction and/or in the conclusion presenting how their findings could be of use to this upcoming application. For example, could these findings be useful for the needed optimization of electrodes applied for water electrolysis?

  • In figures 3, 4 ,5, 7 and 9, the authors depict the formation of metallic nanoparticles and thin films on the Si surface, using a simple sketch. A great addition to this paper would be to complement these illustrations with TEM cross sections images (taken after the electroless depositions), in order to support these assumptions with physical data.

Author Response

Dear reviewer,

Thank you for taking the time to critically review our study. We are pleased that we were able to convince you with our research performance.

According to your comments, we have supplemented our manuscript.

We answered your question about the selection of metals in the text. The metals were selected because they are frequently discussed in the literature. For this purpose, we have included additional literature references. We have also investigated other metal deposits, such as Rh, Ru, and Pd. We would like to discuss these results in the next publication. In this study, we chose to use the species that each have a different charge number (Ag(I), Cu(II), Au(III), Pt(IV)) because that is an essential key to understanding the process for hydrogen formation.

We have also implemented your remark regarding the type of metal coating process in our manuscript. However, our research was focused on the electroless metal deposition because that was the topic of our research project. In the past, many publications have dealt with it, and even today, the topic seems to be up to date.

Although molecular hydrogen formation has been the main focus of our basic research, we have taken up your point about the importance of hydrogen for other research areas. The named subject of chlorate reduction with hydrogen on metal surfaces is a nice transition to the next study we are currently working on.

We like your idea about the TEM images to complement the reaction sketches. Unfortunately, we have not been able to acquire such a device yet. We have at least examined the wafers with SEM images. However, we would like to have the images in the next publication. There are quite interesting correlations between the metals and the surface structuring of the silicon and the hydrogen formation.

Yours sincerely

Stefan Schönekerl and Jörg Acker